# DATA-TO-ENERGY STOCHASTIC DYNAMICS

**Kirill Tamogashev**
University of Edinburgh
k.tamogashev@sms.ed.ac.uk

**Nikolay Malkin**
University of Edinburgh, CIFAR Fellow
nmalkin@ed.ac.uk

 **mmacosha/d2e-stochastic-dynamics**

## ABSTRACT

The Schrödinger bridge problem is concerned with finding a stochastic dynamical system bridging two marginal distributions that minimises a certain transportation cost. This problem, which represents a generalisation of optimal transport to the stochastic case, has received attention due to its connections to diffusion models and flow matching, as well as its applications in the natural sciences. However, all existing algorithms enable the inference of such dynamics only for cases where samples from both distributions are available. In this paper, we propose the first general method for modelling Schrödinger bridges when one (or both) distributions are given by their unnormalised densities, with no access to data samples. Our algorithm relies on a generalisation of the iterative proportional fitting (IPF) procedure to the data-free case, inspired by recent developments in off-policy reinforcement learning for training of diffusion samplers. We demonstrate the efficacy of the proposed *data-to-energy IPF* on synthetic problems, finding that it can successfully learn transports between multimodal distributions. As a secondary consequence of our reinforcement learning formulation, which assumes a fixed time discretisation scheme for the dynamics, we find that existing data-to-data Schrödinger bridge algorithms can be substantially improved by learning the diffusion coefficient of the dynamics. Finally, we apply the newly developed algorithm to the problem of sampling posterior distributions in latent spaces of generative models, thus creating a data-free image-to-image translation method.

## 1 INTRODUCTION

Two modern approaches to generative modelling that have paved the way for scalable and efficient generation of high-fidelity images (Dhariwal & Nichol, 2021; Rombach et al., 2021), videos (Polyak et al., 2024), audio (Chen et al., 2021a; Kong et al., 2021) and text (Nie et al., 2025; Sahoo et al., 2025) are diffusion models and flow matching. Diffusion models (Sohl-Dickstein et al., 2015; Ho et al., 2020; Song et al., 2021b) assume a noising stochastic process that transforms data into a tractable noise distribution and use score-based techniques to learn its reverse process, which transforms noise into data. Flow matching (Liu et al., 2023; Albergo et al., 2025; Lipman et al., 2023; Tong et al., 2024a) learns time-dependent deterministic dynamics that give a transportation map between two arbitrary distributions. Both approaches can be seen as special cases of the more general problem of learning stochastic dynamics between two arbitrary distributions.

The problem of inferring an optimal stochastic bridge between two distributions is called the Schrödinger bridge (SB) problem. Initially proposed in Schrödinger (1931; 1932), it has recently been studied using various machine learning techniques (Huang et al., 2021; Vargas et al., 2021; Chen et al., 2021b; Stromme, 2023; Shi et al., 2023; Tong et al., 2024b). One computational approach to the Schrödinger bridge problem is the iterative proportional fitting (IPF) algorithm (Fortet, 1940; Vargas et al., 2021; De Bortoli et al., 2021), which maintains a pair of processes in forward and reverse time and iteratively updates them by solving half-bridge problems (see §2). Upon convergence, the two processes become time reversals of each other and solve the SB problem. Notably, the typical training of diffusion models – with a fixed noising process that transforms a data distribution into a Gaussian by construction – is a degenerate case of IPF that converges in a single step. However, existing variants of IPF work only in a setting where samples from both marginal distributions are

available and thus cannot be used to model bridges when one (or both) of the marginal distributions is given as an unnormalised density, without access to data samples.

We propose to extend the IPF algorithm to the case where one or both marginal distributions are given by unnormalised densities or energy functions: $p(x) = e^{-\mathcal{E}(x)}/Z$, $Z = \int e^{-\mathcal{E}(x)}\,dx$, where $\mathcal{E}$ can be queried, but $Z$ is unknown. Our proposed *data-to-energy* (or *energy-to-energy*) IPF generalises recently developed techniques for training diffusion models to sample from a distribution given by an unnormalised density (Zhang & Chen, 2022; Vargas et al., 2023; 2024; Berner et al., 2024; Albergo & Vanden-Eijnden, 2025; Blessing et al., 2025a, *inter alia*). In particular, we build upon off-policy reinforcement learning losses and stabilisation techniques for diffusion samplers (Richter et al., 2020; Richter & Berner, 2024; Lahlou et al., 2023; Sendera et al., 2024; Gritsaev et al., 2025) to propose an efficient training for the IPF steps in the data-to-energy setting. Our algorithm is the first general method for inferring data-to-energy and energy-to-energy Schrödinger bridges.

Wielding the newly proposed algorithm, we make three contributions:

(1) We show that the proposed data-to-energy and energy-to-energy IPF algorithms successfully learn stochastic bridges with low transport cost between synthetic datasets and densities, performing on par with the transports learnt by data-to-data IPF using samples from a ground-truth oracle.

(2) As a secondary contribution, we show that – as a consequence of the time discretisation used in our reinforcement learning formulation – existing data-to-data IPF algorithms can be improved by learning the diffusion coefficient of the dynamics, in addition to the drift, generalising the results of Gritsaev et al. (2025) for diffusion samplers to the more general SB setting.

(3) We apply the data-to-energy IPF algorithm to the problem of translating prior distributions to posteriors in latent spaces of generative models, generalising the outsourced diffusion sampling of Venkatraman et al. (2025) to yield a scalable data-free image-to-image translation method.

## 2 DATA-TO-DATA SCHRÖDINGER BRIDGES

### 2.1 ITERATIVE PROPORTIONAL FITTING FOR DATA-TO-DATA SB

**Setting: The SB problem and its connection to optimal transport.** We present some background on the SB problem; see Léonard (2014); De Bortoli et al. (2021) for relevant and more detailed overviews. Let $p_0$ and $p_1$ be two given distributions over the space $\mathbb{R}^d$, assumed to be absolutely continuous (thus used interchangeably with their density functions) and of finite variance. Let $\mathbb{Q}_t$ be a reference process on the time interval $[0, 1]$ taking values in $\mathbb{R}^d$ (usually an Ornstein-Uhlenbeck process, such as the Wiener process). The Schrödinger bridge problem can be formalised as:

$$\mathbb{P}_t^* = \operatorname*{argmin}_{\mathbb{P}_t} \left\{ \mathrm{KL}\left(\mathbb{P}_t \,\|\, \mathbb{Q}_t\right) \text{ s.t. } (\pi_0)_\# \mathbb{P}_t = p_0, (\pi_1)_\# \mathbb{P}_t = p_1 \right\}, \tag{1}$$

where the minimisation is taken over all processes $\mathbb{P}_t$ whose marginals at times $t = 0$ and $t = 1$, written $(\pi_0)_\# \mathbb{P}_t$ and $(\pi_1)_\# \mathbb{P}_t$, equal $p_0$ and $p_1$, respectively. The solution $\mathbb{P}_t^*$ is a stochastic dynamical system that transports $p_0$ to $p_1$ – that is, a *bridge* – that is the closest to $\mathbb{Q}_t$ in KL divergence. If the reference process $\mathbb{Q}_t$ is given by an Itô stochastic differential equation (SDE)

$$\mathbb{Q}_t : \quad dX_t = F_{\mathrm{ref}}(X_t, t)\,dt + \sigma_t\,dW_t, \quad X_0 \sim q_0,$$

then, under mild conditions (see Léonard (2014)) the solution to (1) exists, is unique, and also takes the form of an SDE:

$$\mathbb{P}_t : \quad dX_t = F(X_t, t)\,dt + \sigma_t\,dW_t, \quad X_0 \sim p_0.$$

with the same diffusion coefficient. The KL then takes the form of a dynamic transport cost (Berner et al., 2026)

$$\mathrm{KL}(\mathbb{P}_t \,\|\, \mathbb{Q}_t) = \mathrm{KL}(p_0 \,\|\, q_0) + \mathbb{E}_{X_t \sim \mathbb{P}_t} \int_0^1 \frac{\|F_{\mathrm{ref}}(X_t, t) - F(X_t, t)\|^2}{2\sigma_t^2}\,dt, \tag{2}$$

reducing the problem (1) to one of inferring the drift function $F$ minimising the cost (2). This representation makes clear that as $\sigma_t \to 0$, the SB problem approaches the dynamic optimal transport problem between $p_0$ and $p_1$ with squared-euclidean cost (see Tong et al. (2024a)). For $\sigma_t > 0$, the joint marginal distribution of $\mathbb{P}_t^*$ over $X_0, X_1$ is an *entropy-regularised* optimal transport, a key observation in the derivation of the SB algorithm in Tong et al. (2024b).

**Iterative proportional fitting.** Computationally, the SB problem can be solved using the iterative proportional fitting (IPF) algorithm (Fortet, 1940; Vargas et al., 2021; De Bortoli et al., 2021). IPF defines a recursion initialised at $\overrightarrow{\mathbb{P}}^0_t = \mathbb{Q}_t$:

$$\overleftarrow{\mathbb{P}}^{n+1}_t = \underset{\mathbb{P}_t}{\arg\min} \left\{ \mathrm{KL}\left(\mathbb{P}_t \parallel \overrightarrow{\mathbb{P}}^n_t\right) \text{ s.t. } (\pi_0)_{\#}\mathbb{P}_t = p_0 \right\} \qquad = p_0 \otimes \overrightarrow{\mathbb{P}}^n_{t|0}, \tag{3a}$$

$$\overrightarrow{\mathbb{P}}^{n+1}_t = \underset{\mathbb{P}_t}{\arg\min} \left\{ \mathrm{KL}\left(\mathbb{P}_t \parallel \overleftarrow{\mathbb{P}}^{n+1}_t\right) \text{ s.t. } (\pi_1)_{\#}\mathbb{P}_t = p_1 \right\} \qquad = p_1 \otimes \overleftarrow{\mathbb{P}}^{n+1}_{t|1}, \tag{3b}$$

where each step is the solution to a *half-bridge* problem, which pins the previous iterate at one of the marginals $p_0$ or $p_1$ (here $\overrightarrow{\mathbb{P}}^n_{t|0}$ denotes the conditional process given $X_0$ and $\overleftarrow{\mathbb{P}}^{n+1}_{t|1}$ the conditional process given $X_1$). IPF for Schrödinger bridges is thus a dynamic generalisation of the Sinkhorn algorithm (Sinkhorn, 1964), which computes entropic optimal transport by iteratively renormalising a cost matrix over rows and columns. It can be shown (De Bortoli et al., 2021) that the iterates $\overrightarrow{\mathbb{P}}^n_t$ and $\overleftarrow{\mathbb{P}}^n_t$ converge to the same stochastic process and that this process solves the SB problem (1).

If $\overrightarrow{\mathbb{P}}^0_t = \mathbb{Q}_t$ is represented as an SDE, then the iterates defined in (3b) and (3a) can be represented as forward-time and reverse-time SDEs initialised at $p_0$ and $p_1$, respectively. We can thus introduce SDEs with neurally parametrised drifts (Tzen & Raginsky, 2019):

$$\overrightarrow{\mathbb{P}}^n_t : \quad \mathrm{d}\overrightarrow{X}_t = \overrightarrow{F}_{\theta_n}(X_t, t)\,\mathrm{d}t + \sigma_t\,\mathrm{d}\overrightarrow{W}_t, \tag{4a}$$

$$\overleftarrow{\mathbb{P}}^n_t : \quad \mathrm{d}\overleftarrow{X}_t = \overleftarrow{F}_{\varphi_n}(X_t, t)\,\mathrm{d}t + \sigma_t\,\mathrm{d}\overleftarrow{W}_t, \tag{4b}$$

where $\sigma_t$ coincides with the diffusion coefficient of the reference process, and perform the iterations (3) as optimisation problems over $\theta_n$ and $\varphi_n$. (We henceforth occasionally drop the subscript $n$ from the parameters, with the understanding that (3a) optimises $\varphi = \varphi_{n+1}$ under a fixed $\theta_n$ and (3b) optimises $\theta = \theta_{n+1}$ under a fixed $\varphi_{n+1}$.)

**Data-to-data IPF in a time discretisation.** To approximately perform the optimisations involved in IPF with respect to the parameters $\theta$ and $\varphi$, we discretise the SDEs (4) representing $\overrightarrow{\mathbb{P}}_t$ and $\overleftarrow{\mathbb{P}}_t$ over $K$ steps using the Euler-Maruyama scheme with $\Delta t = \frac{1}{K}$. The continuous-time processes are thus approximated by discrete-time Markov chains, *i.e.*, joint distributions over discrete-time trajectories $\tau = (x_0, x_{\Delta t} \ldots, x_1)$, having the factorisation:

$$\overrightarrow{p}_\theta(\tau) = p_0(x_0) \overbrace{\prod_{k=0}^{K-1} \overrightarrow{p}_\theta(x_{(k+1)\Delta t} \mid x_{k\Delta t})}^{\overrightarrow{p}_\theta(\tau \mid x_0)}, \quad \overrightarrow{p}_\theta(x_{(k+1)\Delta t} \mid x_{k\Delta t}) = \overbrace{\mathcal{N}\left(x_{k\Delta t} + \overrightarrow{F}_\theta(x_{k\Delta t}, k\Delta t)\Delta t, \sigma^2_{k\Delta t}\Delta t\right)}^{\text{Euler-Maruyama step with step size } \Delta t}, \tag{5a}$$

$$\overleftarrow{p}_\varphi(\tau) = p_1(x_1) \overbrace{\prod_{k=1}^{K} \overleftarrow{p}_\varphi(x_{(k-1)\Delta t} \mid x_{k\Delta t})}^{\overleftarrow{p}_\varphi(\tau \mid x_1)}, \quad \overleftarrow{p}_\varphi(x_{(k-1)\Delta t} \mid x_{k\Delta t}) = \underbrace{\mathcal{N}\left(x_{k\Delta t} + \overleftarrow{F}_\varphi(x_{k\Delta t}, k\Delta t)\Delta t, \sigma^2_{k\Delta t}\Delta t\right)}_{\text{reverse-time Euler-Maruyama step with step size } \Delta t}. \tag{5b}$$

As proposed in Vargas et al. (2021), the two IPF optimisation problems in (3) can be approximately solved by maximum likelihood. On the level of the discretised processes, this amounts to the following recurrence:

$$\varphi_{n+1} = \underset{\varphi}{\arg\max}\, \mathbb{E}_{x_0 \sim p_0,\, \tau \sim \overrightarrow{p}_{\theta_n}(\tau \mid x_0)} \sum_{k=1}^{K} \log \overleftarrow{p}_\varphi(x_{(k-1)\Delta t} \mid x_{k\Delta t}) \tag{6a}$$

$$\theta_{n+1} = \underset{\theta}{\arg\max}\, \mathbb{E}_{x_1 \sim p_1,\, \tau \sim \overleftarrow{p}_{\varphi_{n+1}}(\tau \mid x_1)} \sum_{k=0}^{K-1} \log \overrightarrow{p}_\theta(x_{(k+1)\Delta t} \mid x_{k\Delta t}). \tag{6b}$$

The optimisation problem in (6a) (resp. (6b)) requires taking a sample from the marginal distribution $x_0 \sim p_0$ (resp. $x_1 \sim p_1$), rolling out a trajectory in forward time from $\overrightarrow{p}_\theta$ (resp. in reverse time from $\overleftarrow{p}_\varphi$) initialised at the sample, and maximising the log-likelihood of this trajectory in the opposite direction, *i.e.*, under $\overleftarrow{p}_\varphi$ (resp. under $\overrightarrow{p}_\theta$). This procedure essentially requires samples from $p_0$ (resp. from $p_1$) to be available.

**Algorithm 1:** Data-to-Data IPF

1 **for** $n = 0, \ldots, n_{\max}$ **do**
      /* Backward IPF step (6a) */
2   **while** *not converged* **do**
3     $x_0 \sim p_0$
4     $\tau = (x_0, x_{\Delta t}, \ldots, x_1) \sim \overrightarrow{p}_\theta(\tau \mid x_0)$ using (5a)
5     Gradient step on $\varphi$ with $\nabla \log \overleftarrow{p}_\varphi(\tau \mid x_1)$
      /* Forward IPF step (6b) */
6   **while** *not converged* **do**
7     $x_1 \sim p_1$
8     $\tau = (x_0, \ldots, x_{1-\Delta t}, x_1) \sim \overleftarrow{p}_\varphi(\tau \mid x_1)$ using (5b)
9     Gradient step on $\theta$ with $\nabla \log \overrightarrow{p}_\theta(\tau \mid x_0)$
10 **return** $\theta, \varphi$

**Algorithm 2:** Data-to-Energy IPF

1 Initialise buffer $\mathcal{B} = \emptyset$
2 **for** $n = 0, \ldots, n_{\max}$ **do**
      /* Backward IPF step (6a) */
3   **while** *not converged* **do**
4     $x_0 \sim p_0, \tau \sim \overrightarrow{p}_\theta(\tau \mid x_0)$
5     Gradient step on $\varphi$ with $\nabla \log \overleftarrow{p}_\varphi(\tau \mid x_1)$
6     Update $\mathcal{B}$ with samples $x_1$
      /* Forward IPF step (8) */
7   **while** *not converged* **do**
8     **if** *on-policy* **then**
9       $x_0 \sim p_0, \tau \sim \overrightarrow{p}_\theta(\tau \mid x_0)$
10     **else**
11       $x_1 \sim \mathcal{B}, \tau^{(1)} \sim \overleftarrow{p}_\varphi(\tau \mid x_1)$
12       $\tau^{(2)}, \ldots, \tau^{(N)} \sim \overrightarrow{p}_\theta(\tau \mid x_0^{(1)})$
13     Gradient step on $\theta$ with
   $$\nabla \mathrm{Var}_i \left( \log \frac{\overrightarrow{p}_\theta(\tau^{(i)} \mid x_0^{(1)})}{\overleftarrow{p}_\varphi(\tau^{(i)} \mid x_1^{(i)})} + \mathcal{E}_1(x_1^{(i)}) \right)$$
14 **return** $\theta, \varphi$

Figure 1: **Left:** Algorithm for data-to-data IPF. **Right:** Algorithm for data-to-energy IPF, showing the replay buffer with backward trajectory reuse (§3.1), with differences highlighted in red.

We call this iterative algorithm the *log-likelihood method* and give an algorithmic presentation in Algorithm 1. It contrasts with the method proposed in De Bortoli et al. (2021), which uses a slightly different discretisation scheme, although the two coincide in the continuous-time limit ($K \to \infty$).

**Diffusion models as a special case.** Diffusion models (Sohl-Dickstein et al., 2015; Ho et al., 2020; Song et al., 2021b) can be seen as a special case of the IPF algorithm that converges in a single step. If $p_0$ is a data distribution and $p_1$ is Gaussian, and $\mathbb{Q}_t$ is a noising process that transports any source distribution to $p_1$ by construction, then $\mathbb{Q}_t$ initialised at $p_0$ is already a bridge between $p_0$ and $p_1$. Thus the first iteration of IPF – learning $\overleftarrow{\mathbb{P}}_t^1$ by maximum-likelihood training (6a) on trajectories sampled from $p_0 \otimes \mathbb{Q}_t|_0$ – already yields a bridge between $p_0$ and $p_1$, so all subsequent iterations are redundant and $\overleftarrow{\mathbb{P}}_t^1$ solves the SB problem. In practice, training $\overleftarrow{\mathbb{P}}_t^1$ as a neural SDE proceeds by score matching, which is simply a Rao-Blackwellised estimate of the IPF maximum-likelihood objective (Song et al., 2021a).

## 2.2 DISCRETISATION ALLOWS FLEXIBLE KERNELS

The majority of existing work (Chen et al., 2021b; Vargas et al., 2021; De Bortoli et al., 2021; Shi et al., 2023) trains only the drift functions $\overrightarrow{F}_\theta, \overleftarrow{F}_\varphi$ or related objects using objectives similar to (6). In contrast to that, **we propose to train not only the drift, but also diffusion coefficients of both processes**, by replacing the variances $\sigma_{k\Delta t}^2$ in (5) by learnt functions $\overrightarrow{\sigma}_\theta^2(x_k, k\Delta t)$ and $\overleftarrow{\sigma}_\varphi^2(x_k, k\Delta t)$. We expect this to correct for the effect of time discretisation error, inspired by the results for diffusion samplers in Gritsaev et al. (2025).

The optimisation problems in (6) can then be solved with respect to the parameters of both the drift and diffusion coefficients. We compare this approach to those that do not learn the variance in Table 1.

## 3 DATA-FREE SCHRÖDINGER BRIDGES

### 3.1 IPF FOR DATA-TO-ENERGY SB

We now consider the setting where samples from $p_0$ are available, but $p_1$ is given by an unnormalised density $p_1(x) = e^{-\mathcal{E}_1(x)}/Z$ and $Z = \int e^{-\mathcal{E}_1(x)} \mathrm{d}x$ is unknown. In this case, the odd-numbered IPF steps (3a) can be performed (via (6a)), but the even-numbered steps (3b) cannot be done using (6b), as they require samples from $p_1$. Instead, we need an objective that would fit $\overrightarrow{\mathbb{P}}_t^{n+1}$ as a forward-time SDE matching $p_1 \otimes \overleftarrow{\mathbb{P}}_{t|1}^{n+1}$ without samples from $p_1$.

In the time discretisation (5), the IPF step (3b) for $\overrightarrow{\mathbb{P}}_t^{n+1}$ requires enforcing that for every $x_0$, $\overrightarrow{p}_\theta(\tau \mid x_0) \propto \overleftarrow{p}_\varphi(\tau \mid x_1) p_1(x_1)$ over all trajectories $\tau = (x_0, x_{\Delta t}, \dots, x_1)$ starting at $x_0$.

To approximately enforce this proportionality, we introduce a source-conditional variant of the second-moment, or log-variance, loss used for training diffusion samplers (Richter et al. (2020); see Berner et al. (2026) for an overview of related losses and their consistency properties in the continuous-time limit). The idea is to minimise the variance of the log-ratio of the two sides of the proportionality over trajectories $\tau$ sharing the same $x_0$:

$$\mathcal{L}_{\mathrm{LV}}(x_0, \theta) = \mathrm{Var}\left(\log \frac{\overrightarrow{p}_\theta(\tau \mid x_0)}{p_1(x_1)\overleftarrow{p}_\varphi(\tau \mid x_1)}\right) = \mathrm{Var}\left(\sum_{k=1}^{K} \log \frac{\overrightarrow{p}_\theta(x_{k\Delta t} \mid x_{(k-1)\Delta t})}{\overleftarrow{p}_\varphi(x_{(k-1)\Delta t} \mid x_{k\Delta t})} + \mathcal{E}_1(x_1)\right), \quad (7)$$

where the variance is taken over some full-support distribution $p^{\mathrm{train}}(\cdot \mid x_0)$ over trajectories. (The normalising constant $Z$ does not affect the variance, nor do we need to know the marginal of the process being learnt at time 0.) The empirical variance over a batch of $N$ trajectories sampled from $p^{\mathrm{train}}(\tau \mid x_0)$ is an unbiased estimate of (7) and can be used in the loss. (The training policy we use below takes $N$ *non-i.i.d.* trajectories, thus $p^{\mathrm{train}}(\cdot \mid x_0)$ can be a distribution over *batches* of $\tau$.)

Because this proportionality must hold for every $x_0$, we must also average (7) over $x_0 \sim p_0^{\mathrm{train}}(x_0)$, where $p_0^{\mathrm{train}}$ is some training distribution over $x_0$. Thus the full objective for the forward IPF step is

$$\mathcal{L}(\theta) = \mathbb{E}_{x_0 \sim p_0^{\mathrm{train}}(x_0)} \mathbb{E}_{\tau^{(1)}, \dots, \tau^{(N)} \sim p^{\mathrm{train}}(\cdot \mid x_0)} \left[\mathrm{Var}_i \left(\log \frac{\overrightarrow{p}_\theta(\tau^{(i)} \mid x_0)}{\overleftarrow{p}_\varphi(\tau^{(i)} \mid x_1^{(i)})} + \mathcal{E}_1(x_1^{(i)})\right)\right]. \quad (8)$$

The choice of $p_0^{\mathrm{train}}(x_0)$ and $p^{\mathrm{train}}(\cdot \mid x_0)$ is very important and we discuss it in the next subsection.

**Comparison with diffusion samplers.** The objective (7) is a conditional variant of the log-variance, or *VarGrad*, loss (Richter et al., 2020) previously used for diffusion-based samplers of unnormalised target densities. In that setting – *but not in ours*, since the marginal of $p_1 \otimes \overleftarrow{\mathbb{P}}_{t|1}^{n+1}$ at time 0 is not $p_0$ unless IPF has converged – the density of the process being learnt at time 0 is known to be $p_0(x_0)$. Thus $p_0(x_0)$ can be placed in the numerator of the loss in (7) and variance can be taken over both $x_0$ and $\tau$. An alternative objective would fit the density at $x_0$, yielding a variant of the related *trajectory balance* (TB) loss for diffusion samplers (Malkin et al., 2022; Lahlou et al., 2023).

Yet another alternative would avoid IPF altogether and simply perform joint optimisation of both $\theta$ and $\varphi$ using a VarGrad- or TB-like objective to enforce that $p_0(x_0)\overrightarrow{p}_\theta(\tau \mid x_0) = p_1(x_1)\overleftarrow{p}_\varphi(\tau \mid x_1)$ over all trajectories $\tau$. Such a *bridge sampling* approach is taken in Blessing et al. (2025a;b); Gritsaev et al. (2025). However, this approach would yield a bridge that is not necessarily the solution to the SB problem (1), since the KL to the reference process is not minimised. In a future work, it would be interesting to compare our IPF approach with those that regularise bridge sampling losses by the cost (2).

### 3.2 OFF-POLICY TRAINING METHODS

The objective (8) leaves room for the choice of training distributions $p_0^{\mathrm{train}}(x_0)$ and $p^{\mathrm{train}}(\tau \mid x_0)$, which can vary over the course of training. In this way, training with this loss is a form of off-policy reinforcement learning, a connection that has been elucidated and exploited for improved training of diffusion samplers in Sendera et al. (2024).

A naïve choice would take $p_0^{\mathrm{train}} = p_0$ and $p^{\mathrm{train}}(\tau \mid x_0) = \overrightarrow{p}_\theta(\tau \mid x_0)$ (*on-policy* training). However, for complex, high-dimensional distributions on-policy training is insufficient, as modes that are not

discovered by the sampler are very unlikely to be explored. To facilitate training we adapt practices from diffusion sampling literature to guide the sampling process towards the areas of high density of the target distribution $p_1$. In the following paragraphs, we discuss such techniques.

**Replay buffer.** We keep a replay buffer of final samples $x_1$ from the process $\overrightarrow{p}_\theta(\tau \mid x_0)$. During training, to obtain $x_0$, we sample $x_1$ from the buffer, then sample a reverse trajectory to obtain $x_0$ for training: $\tilde{x}_0 \sim \overleftarrow{p}_\varphi(\cdot \mid x_1)$. As the model trains and begins to better approximate $p_1$, the buffer becomes populated with samples $x_1$ that are probable under $p_1$. Thus, the buffer helps the sampler focus on the relevant regions of the space and retain information about previously discovered modes.

**Reverse trajectories.** To obtain the batch of trajectories $\tau$ starting at $x_0$, we use both the reverse trajectory used to produce $x_0$ (see above) and a batch of $N - 1$ trajectories drawn on-policy from $\overrightarrow{p}_\theta(\tau \mid x_0)$ to form a batch of $N$ trajectories sharing their initial point. The reuse of the backward trajectories allows the algorithm to learn on trajectories that reach high-density regions of $p_1$. However, using *only* reverse trajectories prevents the model from sufficiently exploring the whole space. Therefore, careful tuning is needed to strike a perfect balance between exploration and exploitation. We use $N = 2$ for all our experiments.

**Langevin updates.** Following the method of Sendera et al. (2024) for diffusion samplers, we periodically update the buffer using a few steps of unadjusted Langevin on the density $p_1$ to correct for the sampler's imperfect fit to the target.

**Mixing on-policy and off-policy training.** In the training policy, we use a mixture of initial points $x_0$ and trajectories $\tau$ sampled on-policy and those sampled using the Langevin-updated buffer with reverse trajectory reuse, as described above. The frequency of using the buffer is called the *off-policy ratio*, and we ablate different off-policy ratios in Table 3. For most of the experiments we use constant off-policy ratio 0.8.

We provide details of all off-policy methods in Appendix D, and these methods are ablated in §5.4.

### 3.3 Iterative proportional fitting with data-to-energy steps

The full IPF algorithm for data-to-energy SB alternates between backward steps that train $\varphi$ to convergence using the maximum-likelihood objective (6a) and forward steps that train $\theta$ to convergence using (8). The backward step is trained using samples from $p_0$ and forward trajectories from $\overrightarrow{p}_\theta(\tau \mid x_0)$, while the forward step is trained using trajectories obtained using the off-policy methods described above. The complete algorithm is summarised in Algorithm 2. We reuse the model weights and preserve the buffer state from the previous IPF step, although we randomly reinitialise a fraction of samples stored in buffer for the outsourced experiments.

**Energy-to-energy generalisation.** The data-to-energy IPF algorithm can easily be generalised to the case where samples from neither $p_0$ nor $p_1$ are available, but both are given by unnormalised densities $p_i(x) = e^{-\mathcal{E}_i(x)}/Z_i$. In this case, both the backward and forward IPF steps must be performed using the variance-based loss (7), with appropriate choices of training distributions (such as keeping separate replay buffers for both marginals). We call this *energy-to-energy* SB and show preliminary results in §5.2.

**Evaluation metrics.** We consider three metrics for evaluating the approximate solutions to the SB problem yielded by our data-to-energy IPF algorithm. Because SB is a constrained optimisation problem, it is necessary to measure both the constraint satisfaction (*i.e.*, that the solution is a transport from $p_0$ to $p_1$) and the cost (divergence from the reference process). To this end we measure ELBO, path KL, and Wasserstein distance to oracle samples from the target distribution (when available); see Appendix B for details.

## 4 Outsourced sampling with Schrödinger bridges

We describe how our algorithm for solving data-to-energy Schrödinger bridges can be applied to the problem of Bayesian posterior sampling under a pretrained generative model prior $p(x)$ by pulling the sampling problem back to its latent space.

Table 1: Comparison of data-to-data IPF methods. **Bold** indicates the best-performing method.

| Distributions → | Gauss ↔ GMM | | Gauss ↔ Two Moons | | Two Moons ↔ GMM | |
|---|---|---|---|---|---|---|
| Algorithm ↓ Metric → | $\mathcal{W}_2^2(\downarrow)$ | Path KL $(\downarrow)$ | $\mathcal{W}_2^2(\downarrow)$ | Path KL $(\downarrow)$ | $\mathcal{W}_2^2(\downarrow)$ | Path KL $(\downarrow)$ |
| DSBM-IMF (Shi et al., 2023) | $0.046_{\pm0.004}$ | $1.004_{\pm0.137}$ | $0.061_{\pm0.034}$ | $\mathbf{1.813_{\pm0.960}}$ | $0.043_{\pm0.012}$ | $\mathbf{1.893_{\pm0.369}}$ |
| DSBM-IMF+ (Shi et al., 2023) | $0.060_{\pm0.049}$ | $\mathbf{0.909_{\pm0.426}}$ | $0.049_{\pm0.007}$ | $1.848_{\pm0.285}$ | $0.038_{\pm0.010}$ | $\mathbf{1.894_{\pm0.324}}$ |
| [SF]²M (Tong et al., 2024b) | $0.041_{\pm0.021}$ | $2.295_{\pm0.513}$ | $0.053_{\pm0.022}$ | $3.321_{\pm1.862}$ | $0.057_{\pm0.030}$ | $3.823_{\pm0.314}$ |
| *IPF-based* | | | | | | |
| DSB mean (De Bortoli et al., 2021) | $0.093_{\pm0.096}$ | $5.886_{\pm4.460}$ | $0.111_{\pm0.041}$ | $6.398_{\pm1.949}$ | $0.078_{\pm0.029}$ | $5.520_{\pm3.163}$ |
| DSB score (De Bortoli et al., 2021) | $0.052_{\pm0.018}$ | $5.645_{\pm3.474}$ | $0.171_{\pm0.149}$ | $14.346_{\pm8.776}$ | $0.066_{\pm0.035}$ | $5.231_{\pm2.802}$ |
| SDE (Chen et al., 2021b) | $\mathbf{0.037_{\pm0.010}}$ | $2.088_{\pm1.228}$ | $0.033_{\pm0.004}$ | $2.262_{\pm0.268}$ | $\mathbf{0.025_{\pm0.010}}$ | $3.915_{\pm0.285}$ |
| LL fixed var. ≈ Vargas et al. (2021) | $\mathbf{0.037_{\pm0.014}}$ | $2.507_{\pm0.366}$ | $0.033_{\pm0.005}$ | $2.351_{\pm0.149}$ | $0.031_{\pm0.011}$ | $3.710_{\pm0.332}$ |
| LL learnt var. (ours) | $\mathbf{0.042_{\pm0.018}}$ | $2.840_{\pm0.668}$ | $\mathbf{0.022_{\pm0.009}}$ | $4.288_{\pm1.876}$ | $\mathbf{0.023_{\pm0.013}}$ | $3.938_{\pm0.526}$ |

Consider a posterior of the form $p(x \mid y) \propto p(x)r(x, y)$, where $p(x)$ is a prior over the data space (*e.g.*, images) and $r(x, y)$ is a constraint function that encodes the conditional information about the sample $x$ (*e.g.*, a class likelihood or match to a text prompt). If the pretrained generative model is expressed as a deterministic function $f$ of a random noise variable $z \sim p(z)$, Venkatraman et al. (2025) proposed to sample the posterior pulled back to the noise space, with density $p(z \mid y) \propto p(z)r(f(z), y)$, using a diffusion sampler. If $z$ is distributed with this density, then samples $f(z)$ follow the desired posterior distribution $p(x \mid y)$ in data space. Such a method was successfully applied in the latent spaces of various models types, such as GANs, continuous normalising flows.

Instead of using a diffusion sampler, we propose to model a Schrödinger bridge between the distributions $p(z)$ and $p(z \mid y)$. Since neither the normalising constant nor samples from the latter distribution are available, we use the data-to-energy algorithm described in §3.1. Modelling a Schrödinger bridge instead of simply a diffusion sampler has the advantage of transporting prior samples to nearby posterior samples in latent space, which is expected to preserve semantic content that is not constrained by $y$; we show this empirically in §5.3.

**Metrics for outsourced stochastic transport.** In order to evaluate the performance of our method on the stochastic optimal transport task in the latent space, we compute path KL in the latent space, as well as the $L^2$ static transport cost between prior samples from $p_0(x_0)$ and the pushed-forward samples from the approximated posterior $\overrightarrow{p}(\tau \mid x_0)p_0(x_0)$. For image tasks with a classifier reward, in order to evaluate the quality of generated images with respect to the target posterior, we use the mean log-constraint value and FID (Heusel et al., 2017). The latter is computed between images decoded from latents sampled from the trained SB model and images of the target class(es), which are not available to the model during training.

## 5 EXPERIMENTS

### 5.1 BENEFITS OF TRAINABLE VARIANCE

In order to show the benefits of trainable variance in time-discretised processes, we present the comparisons of existing data-to-data SB methods with our proposed algorithms, including both learnt-variance and fixed-variance alternatives. The data-to-data experiments are conducted on several synthetic 2-dimensional benchmarks (following Shi et al. (2023)): Gauss ↔ GMM, Gauss ↔ Two Moons, Two Moons ↔ GMM (where Gauss is an isotropic Gaussian and GMM is a mixture of eight Gaussians). We compare with Shi et al. (2023) (DSBM and

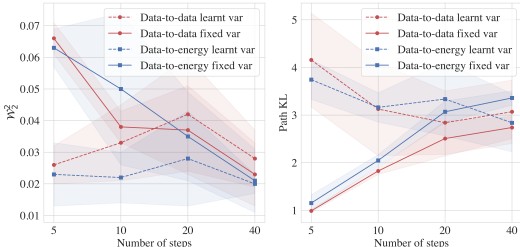

Figure 2: $\mathcal{W}_2^2$ and Path KL depending on the number of discretization steps for Gauss ↔ GMM.

DSBM++), De Bortoli et al. (2021) (DSB), which also uses IPF for training, Chen et al. (2021b) (SDE), which uses a continuous-time version of IPF, and Tong et al. (2024b) ([SF]²M), which relies on a minibatch approximation to entropic optimal transport. All experiments use $K = 20$ discretisation steps. The results in Table 1 clearly show the benefits of training the variances, despite the discrete-time processes not being consistent with an underlying continuous-time process.

We further investigate the effect of learnt variance at varying numbers of discretisation steps. We find that learnt variance allows for more accurate modelling in both data-to-energy and data-to-data settings when the number of steps is small. Results are shown in Fig. 2 (in numerical form in Table 4).

For all 2-dimensional experiments we use $dX_t = \sqrt{2}\, dW_t$ as the reference process; training is done using 4000 steps for both backward and forward processes and 20 IPF steps. We use the same neural network architecture for all 2-dimensional experiments. We provide a detailed experiment configuration in Appendix D.

## 5.2 DATA-TO-ENERGY AND ENERGY-TO-ENERGY SCHRÖDINGER BRIDGE

To prove the viability of data-to-energy Schrödinger bridge training we compare the bridge between Gaussian and GMM distributions trained using data-to-data and data-to-energy IPF versions (where the data-to-energy version has no access to the data samples that the data-to-data algorithm sees). Fig. 3 shows the resulting IPF trajectories and Table 4 shows that the data-to-energy model is comparable to one trained with data samples.

## 5.3 OUTSOURCED SCHRÖDINGER BRIDGE

Finally, we show the scalability of our method by running data-to-energy Schrödinger bridge algorithm in the latent space of generative model. We use the StyleGAN (Karras et al., 2020; 2021) and SN-GAN (Miyato et al., 2018) generators trained on CIFAR-10 (Krizhevsky, 2009), and VAE (Kingma & Welling, 2014; Rezende et al., 2014) generator trained on MNIST (LeCun et al., 1998). We train the bridge model between the latent space prior $p(z)$, which in our case is always a Gaussian distribution, and a reward-reweighted prior of the form $r(f(z), y) \cdot p(z)$. The reward function $r(x, y)$ is a classifier that returns the probability of the object $x$ belonging to the class $y$ (*i.e.*, the probability that $x$ is, for example, a boat for CIFAR-10 or the digit 5 for MNIST).

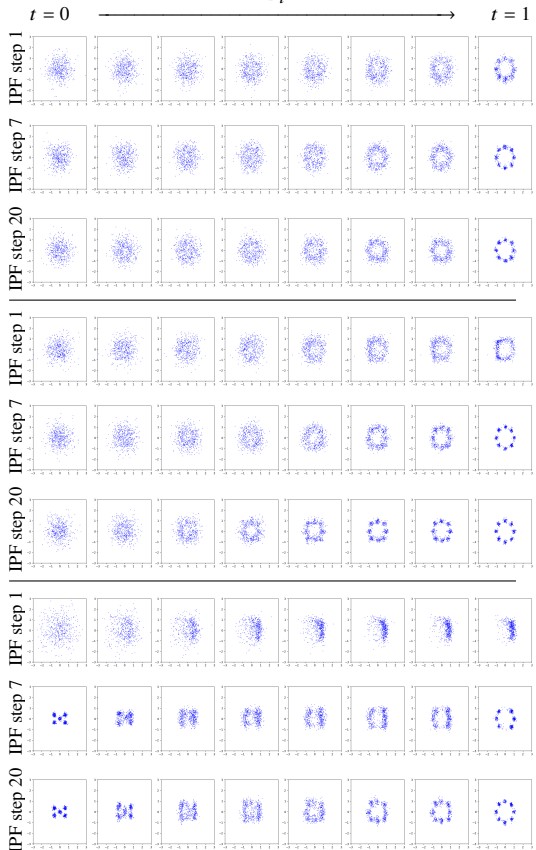

Figure 3: Comparison of learnt processes at various IPF iterations for data-to-data (*top*), data-to-energy (*middle*) and energy-to-energy (*bottom*) settings. For the energy-to-energy setting we use two different mixtures of Gaussian distributions.

MNIST experiments are conducted in two setups: (a) the reward function returns the probability that $x$ is even or odd (b) the reward function returns the probability that $x = 5$. For CIFAR-10 experiments we use StyleGAN (Karras et al., 2020; 2021) generator with latent dimension 512 and SN-GAN (Miyato et al., 2018) generator with latent dimension 128. We use a pretrained classifier model as a reward function. Samples are shown in Fig. 4; more CIFAR-10 results can be found in Appendix F.

In Table 2 we compute FID between samples of the target class posterior obtained from the trained SB and images belonging to the target class in the dataset; we use the same metric for a set of ground truth posterior samples obtained by rejection sampling. Remarkably, the transported samples tend to have lower FID than samples from the true distribution.

These results reveal a benefit of training a Schrödinger bridge model, as opposed to a diffusion sampler or an arbitrary stochastic mapping. It can be seen in Fig. 4 that images already belonging to the target class change little, while those belonging to other classes maintain features that are unrelated to the target class: the background and global structure are preserved. This suggests that style transfer for higher-dimensional images can be a promising application of our method.

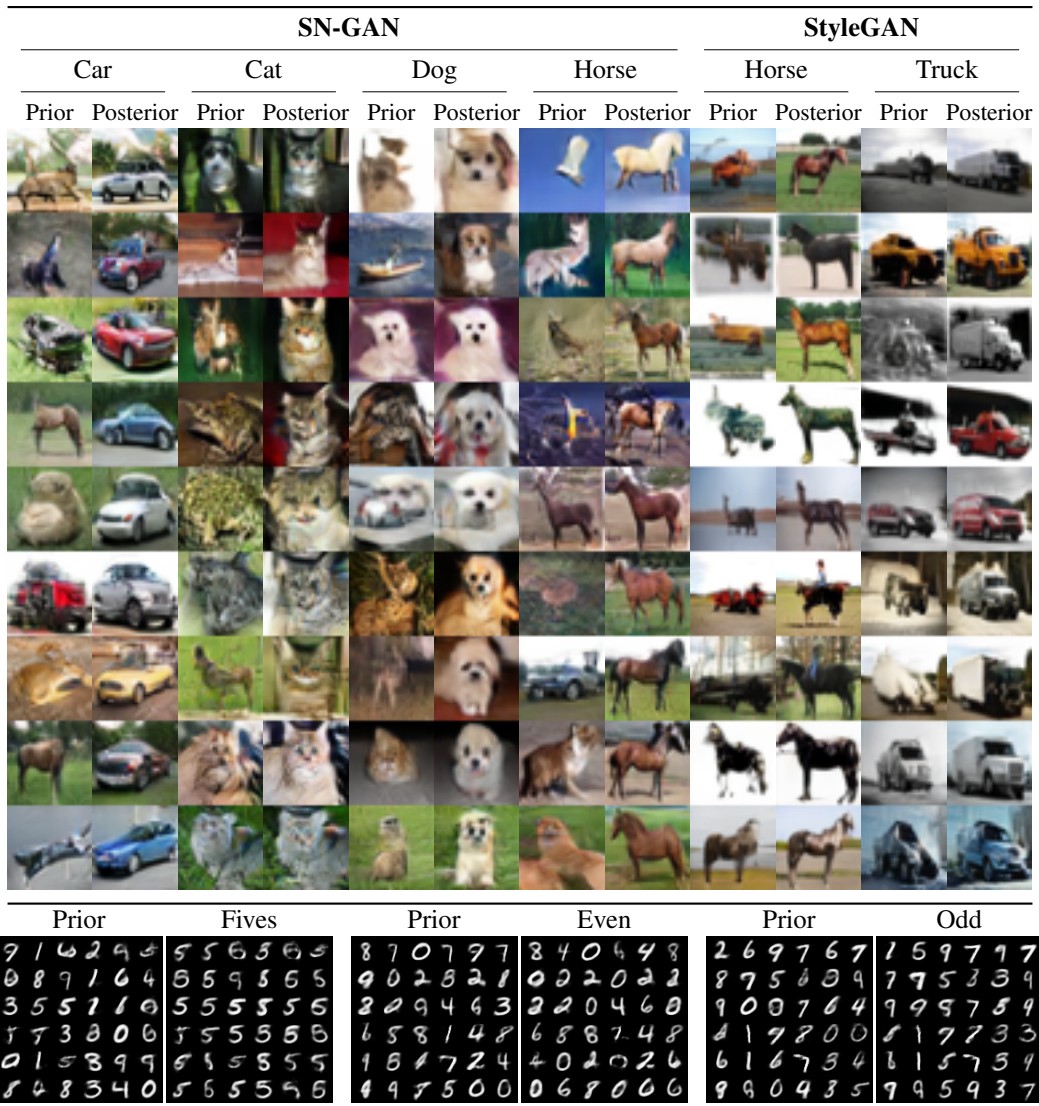

Figure 4: Outsourced Schrödinger bridge on MNIST and CIFAR-10. The bridge preserves style features (thickness, background colour, orientation) while transforming digits to the target class.

Table 2: ELBO (↑) values and FID (↓) scores comparing CIFAR-10 samples from the posterior under a GAN prior and a classifier to data samples of the target class. In this setting diffusion sampler performed worse than other methods, therefore, FID scores are computed only for 'Dog' and 'Car' classes.

| | SN-GAN | | | | StyleGAN | |
|---|---|---|---|---|---|---|
| Algorithm ↓ Class → | Car | Cat | Dog | Horse | Horse | Truck |
| Same class | 10.371 | 17.950 | 15.034 | 12.876 | 12.876 | 9.289 |
| Rejection sampling | 31.334 | 42.242 | 43.691 | 35.019 | 97.855 | 76.403 |
| Langevin | 25.296 | 33.619 | 37.665 | 27.601 | 85.611 | 68.678 |
| Diffusion sampler | 83.940 | - | 60.512 | - | - | - |
| Outsourced SB | 22.312 | 40.489 | 37.287 | 33.021 | 58.988 | 55.346 |
| | ELBO | | | | | |
| Outsourced SB | -6.737 | -9.356 | -6.087 | -5.961 | -21.784 | -6.809 |

Table 3: Ablation of off-policy reinforcement learning techniques on the SN-GAN outsourced sampling problem. **Bold** indicates the best result, underlined indicates second best. The model is trained to amortise sampling from one class (dogs).

| Algorithm ↓ Metric → | ELBO (↑) | Path KL (↓) | $L_2^2(x_0, x_1)(↓)$ | mean log-reward (↑) |
|---|---|---|---|---|
| on-policy | −190.920 | 1506.407 | 10.949 | −0.233 |
| buffer | −188.351 | 622.895 | 10.957 | **−0.125** |
| + Langevin | −188.554 | 383.514 | 10.130 | −0.286 |
| + reuse backward trajectory | **−188.149** | **206.094** | **10.046** | −0.657 |
| + annealed off-policy ratio | −188.355 | 244.270 | 11.386 | −0.149 |
| smaller off-policy ratio | −188.620 | 668.255 | 11.027 | −0.131 |

## 5.4 ABLATIONS OF OFF-POLICY TECHNIQUES

In order to verify our design choices for the high-dimensional experiments we provide a detailed ablation of the various off-policy reinforcement learning tricks described in §3.2. The results are given in Table 3. We use a simple on-policy setup as a baseline. We find that saving samples in a replay buffer (*buffer*) to use them for sampling backward trajectories and applying Langevin update to the buffer samples (*buffer + Langevin*) significantly improves the Path KL metrics, therefore yielding a bridge closer to the optimal one. Moreover, we show that reusing backward trajectories for the computation of loss also improves Path KL, giving the best model based on this metric. However, reusing backward trajectories negatively impacts the mean log-reward metric, possibly because it prevents mode collapse. To balance between modelling modes and achieving low transport cost, we explore the possibilities of both setting a smaller off-policy ratio – fraction of off-policy trajectories – and annealing the off-policy ratio throughout the training; the latter sometimes produces improvements. All ablation experiments are conducted using the SN-GAN generator and VGG13 classifier on the CIFAR-10 dataset. All models are trained to amortise sampling from one class (Dogs) and are trained with the same seed.

## 5.5 COMPARISON WITH ANALYTICAL SOLUTION

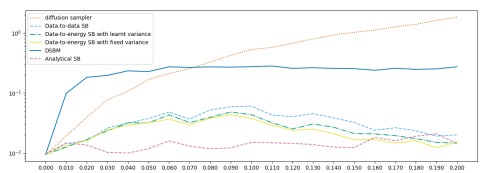

To validate the proposed SB algorithm, we apply it to the problem of finding the SB between two Gaussians. In this case, the problem has an analytical solution (Bunne et al., 2023). For our experiments, we train SB between $\mathcal{N}(0, I)$ and $\mathcal{N}\left(\begin{bmatrix} 1 \\ 1 \end{bmatrix}, \begin{bmatrix} 3.0 & 1.2 \\ 1.2 & 1.0 \end{bmatrix}\right)$. We compare the data-to-energy and data-to-data SB algo-

Figure 5: SB algorithms' comparison in $\mathcal{W}_2$ to the analytical solution.

rithms described in this paper, as well as DSBM (De Bortoli et al., 2021), to the SB computed analytically. For comparison, we use the Wasserstein-2 distance computed between samples obtained from the SB algorithm and analytical solution, for each time step of the trajectory. The results are in Fig. 5.

## 6 CONCLUSION

This paper shows the potential of training data-to-energy Schrödinger bridges in a time discretisation with learnable drift and variance for the forward and backward processes. We showed that, despite its complexity, our method can be successfully scaled. Future work should focus more on scaling the data-to-energy Schrödinger bridges to higher dimensions, as well as arbitrary prior distributions, which would significantly improve the versatility of the proposed algorithms. Moreover, the trained samplers are prone to mode collapse, therefore, future work should investigate techniques to further improve mode coverage. One more promising direction would be to amortise over the distribution of conditions in image generation problems, instead of learning a model for each specific condition. Finally, we are excited to explore various domains in which the proposed algorithms can be applied. Interesting areas include text-conditional image reward fine-tuning of diffusion models and discrete Schrödinger bridge problems (recently explored in Carter et al. (2026)).

ACKNOWLEDGEMENTS

The authors thank Sanghyeok Choi for comments on the manuscript.

This work was enabled by the computational resources of the Edinburgh International Data Facility (EIDF) with funding from the Edinburgh Generative AI Laboratory (GAIL).

NM acknowledges support from the CIFAR Learning in Machines and Brains programme.

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

## A    RELATED WORKS

In this section we establish links between our method and other research directions in the literature.

**Optimal transport.**    Optimal transport is a well-established area of research with a solid theoretical background and scalable applied algorithms. The problem is concerned with finding the optimal transportation map which minimises a given transportation cost. Originally, the problem was proposed by Monge (Monge, 1781). In 20th century this problem was reformulated and generalised by Kantorovich in a series of works including Kantorovich & Rubinshtein (1958); Kantorovich (1960). Since then the problem has been rigorously studied, refer to Peyré et al. (2019); Villani (2008) for a detailed presentation of theory. In a discrete case optimal transport can be solved using Sinkhorn's algorithm (Sinkhorn, 1964; Peyré et al., 2019). Recent works have applied optimal transport to a series of machine learning problems, including image-to-image translation (Korotin et al., 2022), voice conversion (Asadulaev et al., 2024), and super-resolution (Gazdieva et al., 2025).

**Schrödinger bridges.**    The Schrödinger bridge problem is concerned with finding stochastic optimal transport dynamics between two distributions. The problem was originally proposed by Schrödinger (Schrödinger, 1931; 1932). The Schrödinger bridge problem can be seen as a regularised version of dynamic optimal transport (Léonard, 2014) and has interesting connections to optimal control theory (Chen et al., 2021c). Computationally, the problem can be solved using Iterative Proportional Fitting (IPF) algorithm (Fortet, 1940; Deming & Stephan, 1940; Sinkhorn, 1964). De Bortoli et al. (2021); Vargas et al. (2021) proposed the scalable formulation of this method that allows to compute Schrödinger bridge between a pair of distributions given by unbiased samples. Chen et al. (2021b) proposes a continuous-time variant of IPF. Methods distinct from IPF have also been proposed (Shi et al., 2023; Tong et al., 2024b); all of them assume access to samples from the target distribution for an unbiased objective.

**Diffusion samplers.**    Data-to-energy Schrödinger bridge is related to the problem of sampling from an unnormalised density. Diffusion samplers (Zhang & Chen, 2022; Vargas et al., 2023; Richter & Berner, 2024; Berner et al., 2024; Blessing et al., 2025a) represent one of the approaches that solve this problem. Some methods use off-policy reinforcement learning techniques (Lahlou et al., 2023; Sendera et al., 2024) to amortise sampling from intractable density. The theoretical connection among various objectives was established in Berner et al. (2026).

**Outsourced sampling.**    The concept of outsourced diffusion sampling – modelling continuous-time dynamics in latent space for posterior inference under pretrained priors – was proposed in Venkatraman et al. (2025). The work shows that sampling can be efficiently conducted in the latent space of a generator, where the density landscape is smoother.

## B    METRICS FOR DATA-TO-ENERGY SB

First, *if both samples from $p_0$ and the density of $p_0$ are available*, we evaluate the quality of the learnt $\overrightarrow{p}_\theta$ as a sampler of $p_1$ using the evidence lower bound:

$$\text{ELBO} = \mathbb{E}_{x_0 \sim p_0(x_0), \tau \sim \overrightarrow{p}_\theta(\tau|x_0)} \left[ \log \frac{\overleftarrow{p}_\varphi(\tau \mid x_1) \exp(-\mathcal{E}_1(x_1))}{\overrightarrow{p}_\theta(\tau \mid x_0) p_0(x_0)} \right] \le \log Z,$$

which equals the true $\log Z = \log \int \exp(-\mathcal{E}_1(x)) \, dx$ if and only if the processes $\overrightarrow{p}_\theta$ and $\overleftarrow{p}_\varphi$ coincide.

Second, *if samples from $p_1$ are available* (even if not available to the learner during training), we report the 2-Wasserstein distance $\mathcal{W}_2$ between batches of true samples from $p_1(x_1)$ and samples obtained from the learnt model $p_\theta(\tau \mid x_0)$, which measures the discrepancy between the target and modelled marginals.

Third, to approximate the cost, we compute the path KL in discrete time:

$$\mathrm{KL}(p_0 \otimes \overrightarrow{\mathbb{P}}_{t|0} \,\|\, p_0 \otimes \mathbb{Q}_{t|0}) \approx \mathrm{KL}\left(p_0(x_0) \otimes \overrightarrow{p}_\theta(\tau \mid x_0) \,\|\, p_0(x_0) \otimes q(\tau \mid x_0)\right) \tag{9}$$

$$= \mathbb{E}_{x_0 \sim p_0, \tau \sim \overrightarrow{p}_\theta(\tau|x_0)}\left[\log \frac{\overrightarrow{p}_\theta(\tau \mid x_0)}{q(\tau \mid x_0)}\right] \tag{10}$$

$$= \mathbb{E}_{x_0 \sim p_0, \tau \sim \overrightarrow{p}_\theta(\tau|x_0)}\left[\sum_{k=0}^{K-1} \mathrm{KL}(\overrightarrow{p}_\theta(x'_{k+1} \mid x_k) \,\|\, q(x'_{k+1} \mid x_k))\right] \tag{11}$$

where $q(\tau)$ is the time discretisation of the reference process $\mathbb{Q}_t$. The estimator using transition KLs in (11) can be seen to be a Rao-Blackwellised (lower-variance) variant of the estimator in (10), and we use it because the KL can be computed analytically (as all transition kernels are Gaussian). It can be shown (Appendix C) that this path KL is equivalent to path energy as used in Shi et al. (2023).

## C    RELATION BETWEEN PATH KL AND PATH ENERGY

In this section we explain the relation between path KL and path energy (Shi et al., 2023). Assuming that the transition kernels are given by:

$$\overrightarrow{p}_\theta(x'_{k+1} \mid x_k) = \mathcal{N}\left(x_k + v_\theta(x_k, k\Delta t)\Delta t, \sigma^2 \Delta t I\right) \tag{12}$$

$$q(x'_{k+1} \mid x_k) = \mathcal{N}\left(x_k, \sigma^2 \Delta t I\right) \tag{13}$$

where $v_\theta(x_t, t)$ is a learnt drift, $\sigma^2$ is constant and is the same for both $\mathbb{P}_{t|0}$ and $\mathbb{Q}_{t|0}$, time-discrete path KL can be written in the following form:

$$\mathrm{KL}\left(p_0(x_0) \otimes \overrightarrow{p}_\theta(\tau \mid x_0) \,\|\, p_0(x_0) \otimes q(\tau \mid x_0)\right) \tag{14}$$

$$= \mathbb{E}_{x_0 \sim p_0, \tau \sim \overrightarrow{p}_\theta(\tau|x_0)}\left[\log \frac{\overrightarrow{p}_\theta(\tau \mid x_0)}{q(\tau \mid x_0)}\right] = \mathbb{E}_{x_0 \sim p_0, \tau \sim \overrightarrow{p}_\theta(\tau|x_0)}\left[\sum_{k=0}^{K-1} \log \frac{\overrightarrow{p}_\theta(x'_{k+1} \mid x_k)}{q(x'_{k+1} \mid x_k)}\right] \tag{15}$$

$$= \mathbb{E}_{x_0 \sim p_0, \tau \sim \overrightarrow{p}_\theta(\tau|x_0)}\left[\sum_{k=0}^{K-1} \left(\frac{v_\theta(x_k, k\Delta t)}{\sigma^2}(x_{k+1} - x_k) - \frac{\|v_\theta(x_k, k\Delta t)\|^2}{2\sigma^2}\Delta t\right)\right] \tag{16}$$

Given that $x_{k+1} - x_k = v_\theta(x'_k, k\Delta t)\Delta t + \sigma\sqrt{\Delta t}\xi_k$, $\xi_k \sim \mathcal{N}(0, I)$ for $0 \leq k \leq K - 1$ and the $\xi_{k_i}$ are independent, the path KL can be finally written as:

$$\mathrm{KL}\left(p_0(x_0) \otimes \overrightarrow{p}_\theta(\tau \mid x_0) \,\|\, p_0(x_0) \otimes q(\tau \mid x_0)\right) \tag{17}$$

$$= \mathbb{E}_{x_0 \sim p_0, \tau \sim \overrightarrow{p}_\theta(\tau|x_0)}\left[\sum_{k=0}^{K-1} \frac{\|v_\theta(x_k, k\Delta t)\|^2}{2\sigma^2}\Delta t\right] \xrightarrow[\Delta t \to 0]{} \frac{1}{2\sigma^2}\mathbb{E}_{p_0(x_0) \otimes \overrightarrow{p}_\theta(\tau|x_0)}\left[\int \|v_\theta(x_t, t)\|^2 \mathrm{d}t\right] \tag{18}$$

The limit is justified by the Girsanov theorem (Särkkä & Solin, 2019). This yields the path energy used in Shi et al. (2023).

## D    EXPERIMENT DETAILS

### D.1    DATA-TO-DATA EXPERIMENTS

All data-to-data experiments are conducted under the unified setup. For the neural network we use an MLP with 3 hidden layers and 64 neurons in each layer; each layer is followed by a LayerNorm(Ba et al., 2016) and SiLU activation function. All neural networks are trained using AdamW optimiser with learning rate 0.0008. Sampling is done in 20 steps with $t_{\max} = 0.2$ and $\mathrm{d}t = 0.01$. We train forward and backward models for 4000 steps at each IPF iteration and we train each model for 20 IPF iterations. [SF]$^2$M is trained using $160,000$ steps. The metrics are computed using $10,000$ samples from the target distributions and $10,000$ samples obtained from the learnt forward process. All the metrics are averaged over 5 seeds (42, 43, 44, 45, 46).

### D.2 2D DATA-TO-ENERGY EXPERIMENTS

For the data-to-energy experiments we use the same neural networks as for data-to-data experiments. We use 20 steps for sampling with $t_{\max} = 0.8$ and $dt = 0.04$. Neural networks are optimised with AdamW optimiser with learning rate 0.0005. When Langevin update is used, we update buffer samples every 500 steps during training of the forward process. Langevin is used with the step size 0.01 and we do 50 updates each time. We use 2 trajectories from each $x_0$ for the computation of the VarGrad loss. All the metrics are averaged over 5 seeds (42, 43, 44, 45, 46).

### D.3 2D ENERGY-TO-ENERGY EXPERIMENTS

We provide a description of energy-to-energy experiment shown in Fig. 3. We use GMM with 5 modes for the distribution $p_0$ and GMM with 8 modes for distribution $p_1$. For both distributions we rely exclusively on the corresponding log-densities. We do not use samples from either $p_0$ or $p_1$. We keep replay buffers for both densities. We initialise both replay buffers with Gaussian noise at the beginning of training. Since samples are unavailable, we use the objective Equation (7) to learn the forward process and a similar objective:

$$\mathcal{L}_{\mathrm{LV}}(x_1, \varphi) = \mathrm{Var}\left(\log \frac{\overleftarrow{p}_\varphi(\tau \mid x_1)}{\overrightarrow{p}_\theta(\tau \mid x_0) p_0(x_0)}\right) = \mathrm{Var}\left(\sum_{k=1}^{K} \log \frac{\overleftarrow{p}_\varphi(x_{(k-1)\Delta t} \mid x_{k\Delta t})}{\overrightarrow{p}_\theta(x_{k\Delta t} \mid x_{(k-1)\Delta t})} + \mathcal{E}_0(x_0)\right), \quad (19)$$

to learn the backward process.

### D.4 OUTSOURCED SCHRÖDINGER BRIDGE EXPERIMENTS

**Experiments on MNIST.** For MNIST experiments, we use a custom VAE with 3 layers in both the decoder and encoder. We use a custom MNIST classifier as a reward model, which consists of 3 MLP layers. Each layer is followed by the ReLU activation function, except for the last one, which uses a sigmoid. We train the forward and backward networks for $5,000$ steps during each IPF iteration, with 20 IPF iterations in total. All the networks are trained with AdamW optimiser using learning rate 0.0008. We do not use Langevin updates for this experiment, relying only on a replay buffer. We use the same MLPs as in 2D data-to-energy experiment to parameterise the backward and forward drift and variance.

**Experiments on CIFAR-10 with SN-GAN and StyleGAN.** For the CIFAR-10 experiments, we use MLP with 3 hidden layers, which has 256 hidden units for the SN-GAN experiments and 512 for the StyleGAN experiments. We train forward network for 500 steps and backward network for 100 steps during each IPF iteration, for a total of 300 IPF iterations. All the networks are trained with AdamW optimiser using learning rate 0.0005. Langevin updates are made every 500 iterations during the training of the forward network. We run Langevin for 500 steps with initial step size of 0.01 and anneal step size to 0.001 during the updates.

We use 20 steps for sampling with $dt = 0.04$ for the StyleGAN experiments and $dt = 0.005$ for SN-GAN experiments. We use Wiener process, $dX_t = \sqrt{2}dW_t$, as the reference process. All the main experiments are conducted with a replay buffer and Langevin updates, with off-policy ratio of 0.8, and the backward trajectories are reused for computing VarGrad loss.

For the reward model we use VGG (Simonyan & Zisserman, 2015) classifier pretrained on CIFAR-10. The weights are taken from `https://github.com/huyvnphan/PyTorch_CIFAR10`. We use VGG-13 for SN-GAN experiments and VGG-19 for StyleGAN experiments.

For the rejection sampling (ground truth), the FID score is computed between $6,000$ images sampled proportional to the probabilities obtained from classifier and $6,000$ images from the CIFAR-10 dataset. For the outsourced SB the FID score is computed between $6,000$ samples from the learnt model and $6,000$ real CIFAR-10 samples. All scores are computed only on the images of a specific class. All other metrics (path KL, mean log-reward, $L_2^2(x_0, x_1)$, ELBO) are computed using a batch size of 512.

Table 4: SB metrics for varying the number of time discretisation steps in data-to-data and data-to-energy setting with both learnt and fixed variance (Gauss $\leftrightarrow$ GMM).

| Number of steps → | $K = 5$ | | $K = 10$ | | $K = 20$ | | $K = 40$ | |
|---|---|---|---|---|---|---|---|---|
| Algorithm ↓ Metric → | $\mathcal{W}_2^2$ | Path KL | $\mathcal{W}_2^2$ | Path KL | $\mathcal{W}_2^2$ | Path KL | $\mathcal{W}_2^2$ | Path KL |
| Data-to-data learnt var. | $0.026_{\pm 0.006}$ | $4.158_{\pm 0.986}$ | $0.033_{\pm 0.012}$ | $3.127_{\pm 0.981}$ | $0.042_{\pm 0.018}$ | $2.840_{\pm 0.668}$ | $0.028_{\pm 0.009}$ | $3.070_{\pm 0.666}$ |
| Data-to-data fixed var. | $0.066_{\pm 0.005}$ | $0.988_{\pm 0.031}$ | $0.038_{\pm 0.005}$ | $1.825_{\pm 0.073}$ | $0.037_{\pm 0.014}$ | $2.507_{\pm 0.366}$ | $0.023_{\pm 0.010}$ | $2.739_{\pm 0.233}$ |
| Data-to-energy learnt var | $0.023_{\pm 0.010}$ | $3.745_{\pm 0.415}$ | $0.022_{\pm 0.008}$ | $3.162_{\pm 0.308}$ | $0.028_{\pm 0.015}$ | $3.337_{\pm 0.775}$ | $0.020_{\pm 0.003}$ | $2.838_{\pm 0.609}$ |
| Data-to-energy fixed var | $0.063_{\pm 0.006}$ | $1.152_{\pm 0.172}$ | $0.050_{\pm 0.024}$ | $2.047_{\pm 0.123}$ | $0.035_{\pm 0.014}$ | $3.069_{\pm 0.226}$ | $0.021_{\pm 0.010}$ | $3.360_{\pm 0.150}$ |

## E  ADDITIONAL RESULTS

In addition to Fig. 2 we also provide the metrics for the ablation in Table 4. We use the same architecture and hyperparameters as in data-to-data experiments and vary only the number of sampling steps. For the data-to-energy runs we use a replay buffer with Langevin updates, the off-policy ratio is set to 0.8 and the backward trajectories are not reused for the loss computation. $\mathcal{W}_2^2$ is computed using $10,000$ ground truth samples and samples from $\overrightarrow{p}(\tau \mid x_0)p_0(x_0)$. Path KL is also computed on $10,000$ samples.

## F  VISUAL EXAMPLES FOR OUTSOURCED SB

### F.1  UNCURATED EXAMPLES

Prior            Posterior

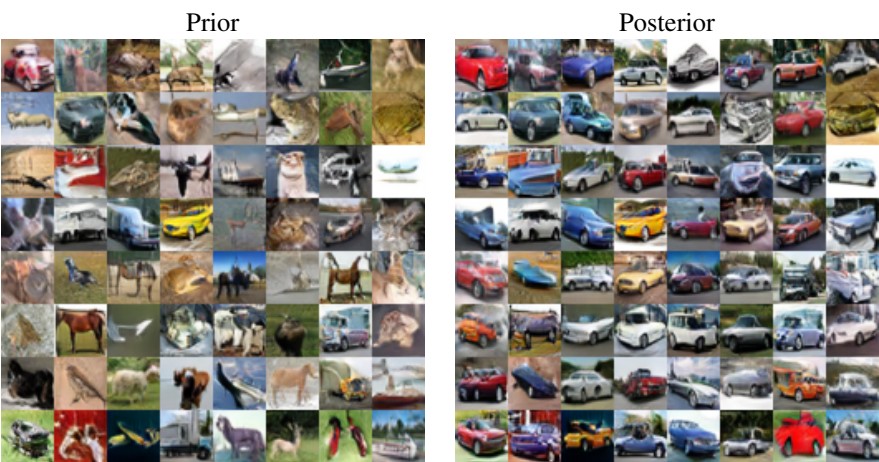

Figure 6: Uncurated examples of outsourced SB with SN-GAN for the class *cars*.

Prior                                          Posterior

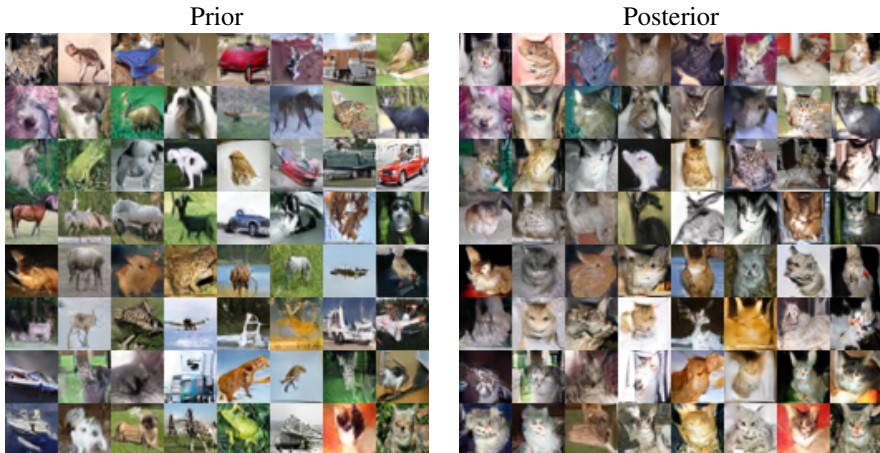

Figure 7: Uncurated examples of outsourced SB with SN-GAN for the class *cats*.

Prior                                          Posterior

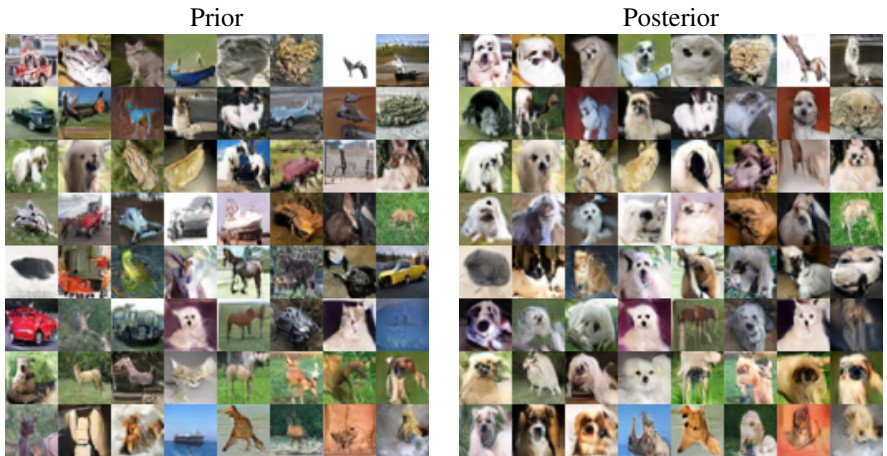

Figure 8: Uncurated examples of outsourced SB with SN-GAN for the class *dogs*.

Prior                                          Posterior

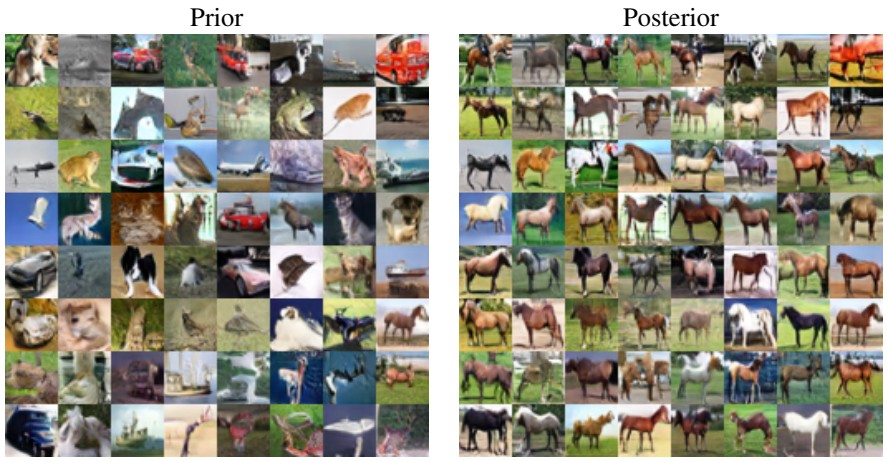

Figure 9: Uncurated examples of outsourced SB with SN-GAN for the class *horses*.

Prior         Posterior

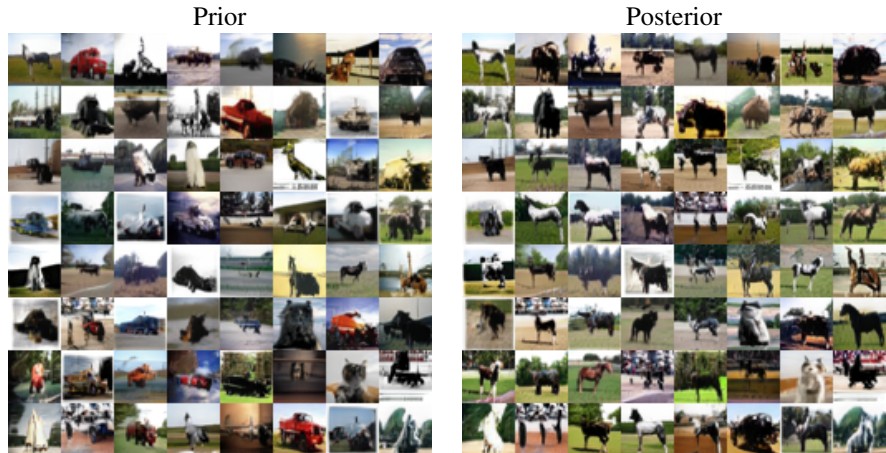

Figure 10: Uncurated examples of outsourced SB with StyleGAN for the class *horses*.

Prior         Posterior

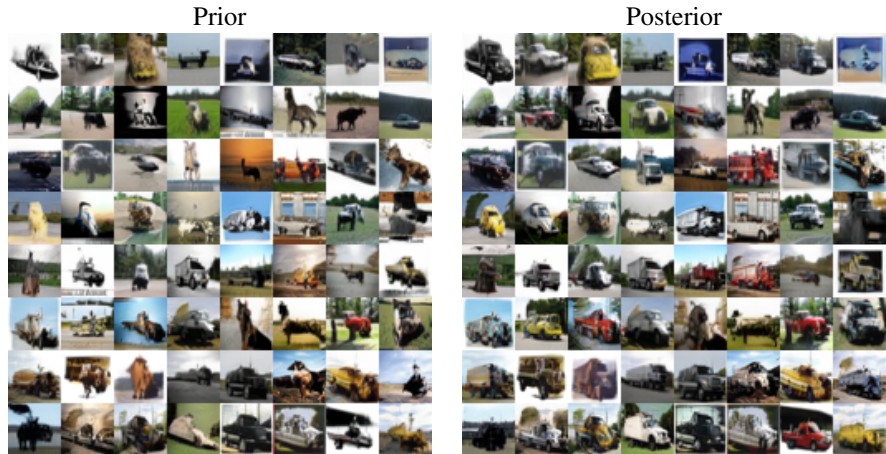

Figure 11: Uncurated examples of outsourced SB with StyleGAN for the class *trucks*.

Posterior (Cats)         Posterior (Dogs)

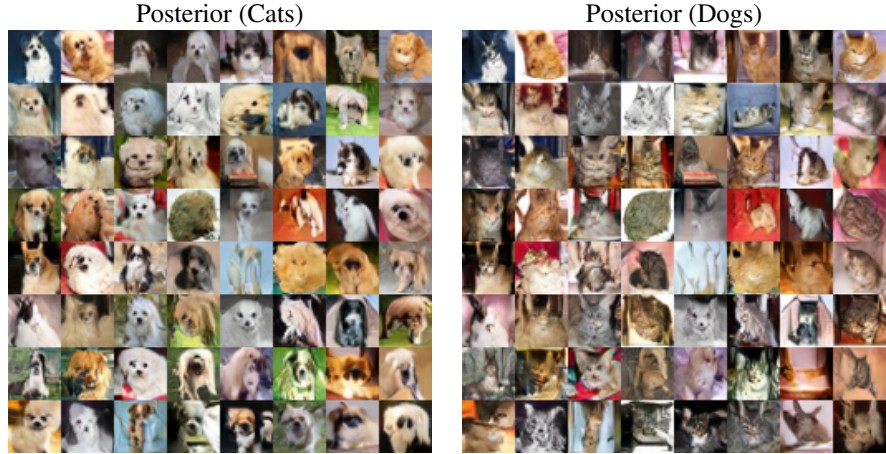

Figure 12: Uncurated examples of outsourced SB with SN-GAN between classes *cats* and *dogs*.

Posterior (Cars)          Posterior (Trucks)

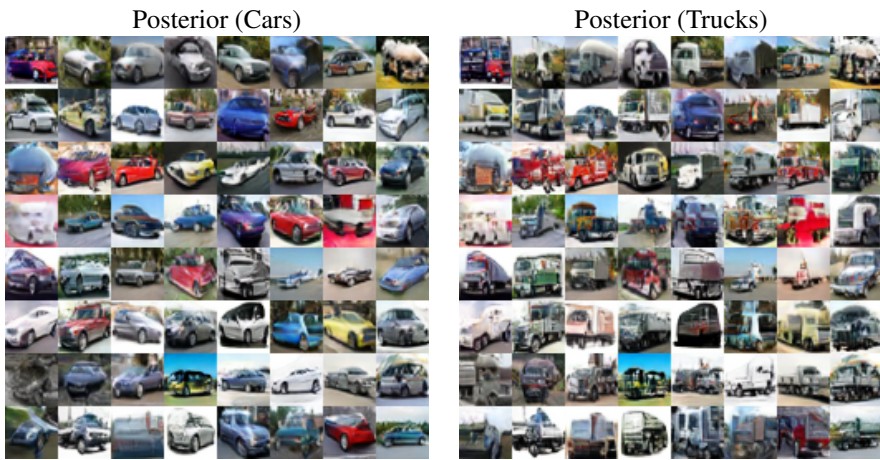

Figure 13: Uncurated examples of outsourced SB with SN-GAN between classes *cars* and *trucks*.

