# OpenReview forum: "Data-to-Energy Stochastic Dynamics"
_ICLR.cc/2026/Conference — ICLR 2026 Poster_

### Official Review · Reviewer_oxUm · 2025-10-29

**Soundness:** 2
**Presentation:** 2
**Contribution:** 2
**Rating:** 4
**Confidence:** 3

**Summary:**

This paper proposes a method for modeling SB when distributions are unnormalized, without accessing samples. This is generally called "data-free," and the proposed data-to-energy IPF procedure attempts to successfully learn transportation between multimodal distributions. The authors validate the approach in various circumstances, including high-dimensional image-to-image translation.

**Strengths:**

* This method is (partially) data-free using an energy-based interpretation; thus, it is likely to find considerable applications in restricted domains.
* The proposed scheme (Eqs. (7-8)) appears to be sound.
* This method can be efficiently outsourced, reaching sufficient scalability for image-to-image translation by using a pretrained latent model (possibly GANs).

**Weaknesses:**

* Limited scalability. The promise of scalability heavily relies on projected and cleaned modalities of embeddings. In other words, the model cannot be a head-first approach to solve an unknown problem, since this requires an unconstrained pretrained generative model.
* Limited comparison. Meaningful comparison studies are only present in synthetic datasets, with mixed results. I can find results from the references indicating that DSB variants show exceptional FID scores, often surpassing those of GANs. The authors are encouraged to report these scores, or acknowledge that their outsourced techniques share limitations on performance with pretrained GANs.
* Contribution. I understand the proposed method as a better-executed IPF with enhanced alignment with the model space. That being said, the manuscript does not successfully demonstrate the benefits of data-to-energy (or energy-to-energy) IPF, and the suggested numerical results contain a lot of overlapping regions, limiting the significance of the methodology.

**Questions:**

* I assume that the training time and memory requirement of Data-to-Energy IPF in synthetic (<50 dim) settings are good, but that it will suffer from the curse of dimensionality for image data (>1000 dim). Could you please give us the training time and memory requirements of standard and Data-to-Energy IPF procedures for various dimensions?

---

> ### Author Response · Authors · 2025-11-26
>
> Thank you for the review, and in particular for noting the soundness and scalability of the proposed algorithms. We will now address the stated weakness and questions:
>
> ### On the problem being solved and our contribution (W3)
>
> As a general comment, let us repeat that the main purpose of this paper is to present **the first general algorithm for Schrödinger bridges between two distributions in the case where data samples are not available from both of them**. This problem has not been solved outside of very restricted cases in the past.
>
> Because the general data-to-energy and energy-to-energy SB problems **have not been considered before**, there are no established datasets specifically for these problems. Therefore, for basic validation of the algorithm, we took examples from existing work on data-to-data SB where a density function happens to be available (for example, Gaussian mixtures).
>
> The data-to-data and data-to-energy algorithms solve different problems, with different data available to the learner, and are not meant to be comparable. (However, notably, we find that data-to-energy algorithms learn a better bridge than data-to-data ones, despite having the former having no access to samples from the target distribution.)
>
> ### Limited scalability (W1)
>
> > The promise of scalability heavily relies on projected and cleaned modalities of embeddings.
>
> We, unfortunately, do not understand this question. What embeddings and pretrained model are meant here?
>
> The only pretrained models we use are the GANs used in image-to-image experiments, but no projection/embedding of points in the data space or "cleaning" occurs -- we never need to obtain latents from data samples.
>
> If we have misunderstood, we would appreciate clarification of the question.

---

> ### Author Response · Authors · 2025-11-26
>
> ### Limited comparisons (W2)
>
> We want to point out the differences between fitting a Schrödinger as an image generative model (as in the works you mention) and the problem we study here. Those models study a data-to-data problem: between samples from a Gaussian to a dataset of images. We, on the other hand, study bridges where the target distribution does not have available data samples. Image generative modelling is not relevant and those algorithms are not comparable (please also see the response to W3 above). Conversely, those existing algorithms (e.g., DSB variants or other data-to-data SB algorithms) are not applicable to the problem of, for example, sampling from a posterior under a pretrained GAN prior with access only to the likelihood (classifier).
>
> In light of the above, the FID scores in the image-to-image experiment should not be compared with FIDs for image generative modelling. They are not FIDs for unconditional image generation (the problem solved by many kinds of image generative models, including ones based on data-to-data SB), but rather FIDs computed between **ground truth images of the target class** (which are *not available to the learner*) and **decoded samples from the trained SB model in the latent space**.
>
> Because we are performing translation in the **latent space** of a GAN without data, evaluation of a sampler against **images** of the target class will incur bias from two sources:
> - from an imperfect prior/decoder: the distribution of decoded prior samples does not match the distribution of real images
> - from an imperfect classifier: the distribution of images weighted by the classifier's probability of a class does not match the distribution of real images of that class.
>
> To better place our FID results in context, for the outsourced Schrödinger bridge experiment, we compute FID between a set of ground truth images of the target class and:
> - an independent sample of the same size of images of the same class (which would give 0 in the limit of infinite sample size);
> - samples from our trained Schrödinger bridge model;
> - samples from a diffusion sampler in latent space that aims to sample the same target distribution as the bridge; this follows [Venkatraman et al., "Outsourced diffusion sampling...", ICML'25];
> - decoded samples from the *exact* posterior in latent space, obtained via rejection sampling;
> - samples obtained by Langevin dynamics in the latent space, again asymptotically sampling from the same target density.
>
> **FID score for SN-GAN trained on CIFAR-10.**
>
> (Some results are still running and will be updated here by 27 November AOE.)
>
> |     Algorithm    |  Car   |  Cat   |  Dog   |  Horse |
> | :--------------: | :----: | :----: | :----: | :----: |
> |Same class        | 10.371 | 17.950 | 15.034 | 12.876 |
> |Rejection Sampling| 31.334 | 42.242 | 43.691 | 35.019 |
> |Langevin          | 40.302 | 45.949 | 42.736 | 34.873 |
> |Diffusion Sampler | 83.940 |  53.253   | 60.512 |   57.251   |
> |Outsourced SB     | 22.312	| 40.489 | 37.287 | 33.021 |
>
>
> **FID score for StyleGAN trained on CIFAR-10.**
>
> |     Algorithm    | Horse  | Truck  |
> | :--------------: | :----: | :----: |
> |Same class        | 12.876 |  9.289 |
> |Rejection Sampling| 97.855 | 76.403 |
> |Langevin          | 85.611 | 68.678 |
> |Outsourced SB     | 58.988	| 55.346 |
>
> **Among amortised and approximate sampling methods (Langevin, diffusion sampler, our SB), the SB achieves the lowest FID.**

---

> ### Author Response · Authors · 2025-11-26
>
> ### Training time (Q1)
>
> First, we recall once again (see W3 above) that data-to-energy SB algorithms are not competitors to data-to-data SB algorithms such as data-to-data IPF. However, we agree that a time analysis can be informative and provide it below for the large-scale problems we consider.
>
> All estimations are conducted using NVIDIA A100-SXM4-80GB GPU. We measure time for reward estimation, Langevin dynamics on the target density, NLL-based update steps, and VarGrad-based update steps. Recall that:
> - data-to-data IPF uses NLL-based IPF steps at all iterations;
> - our proposed data-to-energy IPF alternates between NLL-based and VarGrad-based IPF steps;
> - our proposed energy-to-energy IPF uses VarGrad-based IPF steps at all iterations.
>
> | Models | Reward estimation | Langevin (100 steps) | VarGrad (100 steps) | NLL (100 steps) |
> | :--:     | :--------: | :----------: | :---------: | :------: |
> | StyleGAN | 88.1 ms | 109s | 177s | 27 s |
> | SN-GAN    | 18.4 ms | 7s  | 58s | 27 s |
>
> While VarGrad is more costly than NLL, as the gradient update uses the full trajectory, this **cannot be avoided when target data samples are not available**: intuitively, one must store an entire trajectory before performing a learning update if the learning signal (evaluation of energy) only comes at the end of the trajectory. On this note, we point out that the SB problem without data samples from the target marginal distribution is at least as hard as training a diffusion sampler (i.e., a diffusion model trained to sample from a distribution with given energy without access to samples). All existing training methods for diffusion samplers require at least simulating full trajectories for training.
>
> **We hope the above has helped put our contributions in context and has positively affected your assessment of the paper. Please do not hesitate to ask any further questions.**

---

### Official Review · Reviewer_k5Aj · 2025-10-31

**Soundness:** 2
**Presentation:** 2
**Contribution:** 2
**Rating:** 4
**Confidence:** 4

**Summary:**

This paper studies the problem of data-to-energy generation on synthetic datasets, aiming to learn mappings between data-driven and energy-based distributions. To achieve this, the authors employ an Iterative Proportional Fitting (IPF) procedure, which is essentially a variant of the Sinkhorn algorithm, to bridge the two distributions through alternating updates. The method is evaluated on several synthetic problems, including posterior sampling in the latent space of generative models, and compared with standard data-to-data approaches.

**Strengths:**

- The paper presents an interesting conceptual perspective by framing the connection between data-driven and energy-driven distributions as a transport problem.

- The manuscript is clearly written and easy to follow, with a straightforward presentation of the core algorithmic idea.

**Weaknesses:**

While the proposed idea is conceptually interesting, I have several concerns regarding its scalability, novelty, and experimental rigor.

- **Computational inefficiency.** The proposed method employs standard IPF training, which is known to be computationally demanding. It requires simulating and storing both forward and backward trajectories, leading to significant memory and runtime costs. Moreover, due to the log-variance computation in Eq. (8), the algorithm must maintain multiple trajectories for each source sample $x_0\sim p_0$, further amplifying the computational overhead. This raises serious questions about the method’s scalability to high-dimensional or large-scale settings.

- **Limited comparison with related methods.** The paper does not include comparisons with existing and more scalable energy-to-energy [2] and data-to-energy [3] approaches. These frameworks perform only forward simulations, efficiently handle trajectory storage, and are generally better suited for practical applications. Although [3] may be considered concurrent work, it was publicly available nearly three months before the submission deadline and should at least be discussed for completeness.

- **Questionable experimental design.** The experimental evaluation is restricted to self-curated synthetic datasets and lacks standardized benchmarks, making it difficult to assess general applicability.

(1) On synthetic datasets, it would be more informative to include samplers capable of energy-to-energy or data-to-energy methods, as these are the most directly comparable baselines.

(2) The posterior sampling from latent formulated as $r(f(x), y) p_0(x)$ appears somewhat artificial and lacks a clear grounding in established probabilistic models. In contrast, a more canonical formulation sampling from energy-tilted data distributions $e^{r(x)/\alpha} p_{data}$ is well studied in the literature. For estimating $p_{data}$, the authors could have leveraged normalizing flows [4], continuous normalizing flows [5], or VAE/diffusion-based ELBO approximations that provide explicit or approximate log-likelihoods. Furthermore, comparison against rejection sampling seems too weak; many stronger training-based or training-free posterior sampling methods could serve as more relevant baselines.

(3) Evaluating the approach on more established and quantitatively meaningful benchmark tasks is required. At a minimum, incorporating comparative analyses against existing samplers on standard sampling problems would provide a more rigorous and convincing validation of the proposed framework.

Overall, the absence of experiments on well-established benchmarks and the lack of comparisons with efficient samplers substantially weaken the empirical claims.

- **Limited methodological novelty.** From a theoretical standpoint, the proposed formulation largely adapts the framework of [1] to the case where the energy function is defined on only one marginal. While this adaptation is logically consistent, it does not appear to introduce a fundamentally new methodological contribution beyond prior work.

References

[1] From Discrete-Time Policies to Continuous-Time Diffusion Samplers.

[2] Sequential Controlled Langevin Dynamics.

[3] Adjoint Schrödinger Bridge Sampler.

[4] Glow: Generative Flow with Invertible 1×1 Convolutions.

[5] Normalizing Flows are Capable Generative Models.

**Questions:**

- How exactly is Langevin noise incorporated into the proposed method? Is it applied only at the final iteration, or added throughout the IPF trajectory?

- Does the method still converge or perform meaningfully without Langevin noise, or is it a necessary component for stability?

---

> ### Author Response · Authors · 2025-11-26
>
> Thank you for your review; we are glad to see that you liked the presentation and the conceptual perspective of sampling via dynamic measure transport.
>
> We would like to address the weaknesses and questions you brought up, as many of them seem to rest on important misunderstandings of our motivations and approach.
>
> In general, let us repeat that the main purpose of this paper is to present the first general algorithm for Schrödinger bridges between two distributions in the case where data samples are not available from both of them. This problem has not been solved outside of very restricted cases in the past.
>
> ### Computational efficiency (W1)
>
> It is true that IPF algorithms require an outer loop and can accumulate error. This is true of IPF in the data-to-data setting as well, which is known since [Vargas et al., "Solving Schrödinger bridges...", Entropy, 2021] and [De Bortoli et al,. "Diffusion Schrödinger bridge...", NeurIPS 2021]. However, while solutions not requiring IPF have been developed in later work for the data-to-data setting (as we discuss in the introduction), there is, as yet, no alternative to our proposed IPF in the data-to-energy setting.
>
> We note that the SB problem without data samples from the target marginal distribution is at least as hard as training a diffusion sampler (i.e., a diffusion model trained to sample from a distribution with given energy without access to samples). All existing training methods for diffusion samplers require at least simulating full trajectories for training. On the other hand, an advantage of off-policy objectives, such as VarGrad, is that they do not require *differentiable simulation* of the process, that is, there is no backpropagation into the SDE integration procedure that generates training trajectories, which results in lower computation cost. These advantages are well-known in the diffusion sampling setting, see, e.g., [Sendera et al., "Improved off-policy training...", NeurIPS 2024].
>
> ### Comparisons (W2)
>
> We disagree with the relevance of comparisons to [2,3].
>
> The paper [2] does not study the general data-to-energy SB problem, but rather the diffusion sampling problem.
>
> Regarding the adjoint SB algorithm [3], this **concurrent** work only studies the case of a Gaussian source distribution. However, we take this chance to discuss that work.
>
> Firstly, this paper appeared on arXiv less than three months before the submission deadline. Per ICLR policy, we should not be required to compare with this work, nor is their code made public.
>
> Secondly, there are notable differences between our approach and the very interesting algorithm proposed there. It appears to us that the adjoint SB algorithm has these appealing properties:
> - it uses a regression objective computed at each intermediate $x_t$, rather than a trajectory-level update, which would lead to lower memory requirements;
> - it does not require an outer IPF loop, which would prevent error accumulation.
>
> On the other hand, our proposed algorithms also have some advantages:
> - they allow off-policy training, which is essential for preventing mode collapse (while [3] requires drawing samples from the current sampler for training);
> - they do not require access to the gradient of the target energy (while [3] uses it to compute the regression target);
> - they are applicable to non-Gaussian source distributions (while [3] considers only Gaussian-to-energy problems).
>
> This discussion will be included in the paper. We will also aim to include comparisons with [3] -- using a reimplementation or their published code, if it becomes available -- in the camera-ready version.

---

> > ### Author Response · Authors · 2025-11-26
> >
> > ### Experimental design (W3)
> >
> > Because the general data-to-energy and energy-to-energy SB problems **have not been considered before**, there are no established datasets specifically for these problems. Therefore, for basic validation of the algorithm, we took examples from existing work on data-to-data SB where a density function happens to be available (for example, Gaussian mixtures).
> >
> > The data-to-data and data-to-energy algorithms solve different problems, with different data available to the learner, and are not meant to be comparable. (However, notably, we find that data-to-energy algorithms learn a better bridge than data-to-data ones, despite the former having no access to samples from the target distribution.)
> >
> > > posterior sampling from latent [...] appears somewhat artificial and lacks a clear grounding in established probabilistic models
> >
> > We respectfully strongly disagree with this claim. As shown in [Venkatraman et al., "Outsourced diffusion sampling...", ICML'25], in a model where samples $x$ are generated by sampling $z\sim p_0$ and emitting $x=f(z)$, sampling from a distribution over latent $z$ with density proportional to $r(f(z),y)p_0(z)$ is a correct way to sample from the posterior with density $p_{\rm data}(x)f(x)$ in the data space. That is, **sampling in this way in latent space is a correct and probabilistically grounded manner of sampling the posterior in data space**. In other words, if $p_{\rm data}$ happens to be a distribution defined in this way (i.e., as the pushforward of $p_0$ by $f$), then sampling $z$ from $r(f(z),y)p_0(z)$ results in $x=f(z)$ being distributed with exactly the density $e^{r(x)}p_{\rm data}(x)$ that you describe.
> >
> > Indeed, $f$ could be a normalising flow, continuous normalising flow, etc. However, an advantage of the latent view is that **the above approach is correct even if $f$ is not invertible or the density of $p_{\rm data}$ cannot be computed**, for example, in the case of a GAN.
> >
> > On a minor note, we are confused about one of the references: [5] is not a paper about continuous normalising flows.
> >
> > > comparison against rejection sampling seems too weak
> >
> > We again disagree with this claim, as **rejection sampling gives unbiased samples from the posterior**! That is, it is the theoretically optimal sampling algorithm.
> >
> > ### Methodological novelty (W4)
> >
> > We respectfully disagree with the relevance of the references and with the claims about novelty.
> >
> > > From a theoretical standpoint, the proposed formulation largely adapts the framework of [1] to the case where the energy function is defined on only one marginal.
> >
> > First, [1] studies the limiting properties of various existing objectives for diffusion sampling. It does not propose any new objectives, nor does it study the SB problem, nor does it study the cases where the energy function is defined on zero or or two marginals.
> >
> > > While this adaptation is logically consistent, it does not appear to introduce a fundamentally new methodological contribution beyond prior work.
> > >
> > Second, generalisation of objectives like VarGrad to the problem of transporting arbitrary distributions, as well as its integration in an IPF loop, are novel, and they require some innovations to make it work, as we show in our ablation studies (Table 3). The application of dynamic measure transport in latent space to image translation is similarly original, and we believe our (primarily methodological) work opens the door to many more interesting applications.
> >
> > ### "Langevin noise" (Q1, Q2)
> >
> > > How exactly is Langevin noise incorporated into the proposed method? Is it applied only at the final iteration, or added throughout the IPF trajectory?
> >
> > We are not sure we understand the question.
> >
> > If you are referring to the noise injected during SDE integration, yes, this is done at every step (discrete-time Gaussian transition kernels are equivalent to Euler-Maruyama integration steps on an underlying SDE).
> >
> > If you are referring to the Langevin buffer used for off-policy training, the buffer only stores the terminal states and does not operate on trajectories.
> >
> > If it is something else, please do clarify, and we will be happy to answer.
> >
> > > Does the method still converge or perform meaningfully without Langevin noise, or is it a necessary component for stability?
> >
> > An ablation on removing the Langevin buffer has already been done in Table 3. The buffer is necessary not for stability, but rather for mode coverage and preventing forgetting of earlier discovered modes -- again generalising findings from the diffusion sampling literature [Sendera et al., "Improved off-policy training...", NeurIPS'24].
> >
> > **Thank you again for your review. We hope that the above clarifications will help you to better assess the contributions of our paper. Please let us know if we can provide any further answers during the discussion period.**

---

### Official Review · Reviewer_Rvjv · 2025-11-01

**Soundness:** 3
**Presentation:** 3
**Contribution:** 2
**Rating:** 6
**Confidence:** 3

**Summary:**

This paper proposes a generalization of the Iterative Proportional Fitting (IPF) procedure for Schrödinger Bridge (SB) estimation to settings where one or both marginals are given only by unnormalized densities (energy functions) rather than by data samples. The proposed data-to-energy IPF leverages a variance-based (VarGrad) objective previously used in diffusion samplers for unnormalized target densities. In addition, the paper shows that learning both drift and diffusion coefficients improves discretized SB optimization compared to drift-only models. Extensive experiments demonstrate the method on synthetic 2D datasets (Gaussian, GMM, Two-Moons) and on latent-space posterior sampling tasks (StyleGAN, SN-GAN, VAE). The approach is further extended to an energy-to-energy SB variant, demonstrating fully sample-free transport between energy-based densities.

**Strengths:**

- The paper introduces a generalization of IPF to the data-free setting, allowing SB learning even when only energy functions are available. This is an important theoretical and practical advance, broadening the applicability of SB to posterior inference and energy-based models.
- The authors provide extensive ablations on trainable variance and other off-policy reinforcement learning techniques, such as replay buffers, Langevin updates, and off-policy ratio.
- The inclusion of learnable variance meaningfully improves discrete SB optimization accuracy, especially at low discretization levels.
- The latent-space posterior sampling experiments demonstrate the practical utility of the proposed method.

**Weaknesses:**

- The convergence guarantee of the proposed data-to-energy IPF (Eq. 7) is not established. While classical IPF has convergence results under certain regularity assumptions [1], it remains unclear whether those extend to the variance-based updates and off-policy sampling. A discussion of theoretical convergence would strengthen the contribution.
- The paper lacks comparison with a closely related concurrent work, the Adjoint Schrödinger Bridge Sampler [2], which also addresses SB estimation when one marginal is given as an unnormalized density. A direct comparison or discussion of methodological differences would clarify novelty.
- In Table 1, the Path KL metric is reported as “larger = better.” However, by definition, the SB objective minimizes KL (P ‖ Q), suggesting that smaller Path KL values should correspond to more optimal bridges. If the smaller Path KL is better, then the reported results appear to indicate a tradeoff: the trainable-variance model improves Wasserstein distance but worsens Path KL.  In this case, the justification for employing trainable variance under the tradeoff is required.

[1] De Bortoli, Valentin, et al. "Diffusion schrödinger bridge with applications to score-based generative modeling." NeurIPS 2019.
[2] Liu, Guan-Horng, et al. "Adjoint Schr\" odinger Bridge Sampler." NeurIPS 2025.

**Questions:**

- In Fig. 3, why does the energy-to-energy experiment use a different data setup (GMM-to-GMM), whereas the data-to-data and data-to-energy experiments use Gaussian-to-GMM?

---

> ### Author Response · Authors · 2025-11-26
>
> Thank you for your review, in particular, for noting that we have made an important theoretical and practical advance, appropriately evaluated design choices, and meaningfully demonstrated a path towards large-scale applications.
>
> ### Convergence guarantees not established (W1)
>
> We indeed do not write about convergence guarantees, but are happy to include them in the revision. We discuss them here now.
>
> The convergence of IPF in discrete time, under regularity assumptions on the source and target distributions, was indeed established by [1]. Those results show that if time reversal is achieved perfectly on each iteration of IPF, then the processes converge in the limit to the SB solution.
>
> However, notably, these results there **do not concern function/distribution approximation**, such as fitting Gaussian transitions with mean and possibly variance computed by neural nets of a certain architecture (which is done by all existing works on neural SB learning). In general, perfect time reversal cannot be achieved with Gaussian transitions outside of very restricted cases, e.g., if both source and target are Gaussian measures, although it should be noted that Gaussian transitions provide better approximations in the continuous-time (SDE) limit.
>
> Therefore, the best result we can hope for is an analogue of the result from [1] for data-to-energy IPF, under identical regularity conditions, without assuming Gaussian transitions. For this, it is sufficient to show that the VarGrad objective, which we use to perform time reversal steps with no data samples available, achieves time reversal. It would then follow that the proposed algorithm, if it reaches a global optimum *in the space of all absolutely continuous transition kernels* on each iteration, gives convergence to the SB solution.
>
> Now, the correctness of VarGrad as used in our algorithms -- that is, that VarGrad objective being 0 is equivalent to time reversal being achieved -- is easily reduced to well-established results such as those from the diffusion sampling literature; see, for example, [Berner et al., "From discrete-time policies...", 2025] for discussion.
>
> We are happy to include the above discussion, to a reasonable level of formality, in the revision.
>
> ### Concurrent work [2] (W2)
>
> Thank you for mentioning this work.
>
> Firstly, we note that while this paper will appear in NeurIPS 2025, it appeared on arXiv less than three months before the submission deadline. Per ICLR policy, we should not be required to compare with this work, nor is their code made public.
>
> Secondly, we take this chance to discuss the differences between our approach and the very interesting algorithm proposed in [2].
>
> It appears to us that the adjoint SB algorithm has these appealing properties:
> - it uses a regression objective computed at each intermediate $x_t$, rather than a trajectory-level update, which would lead to lower memory requirements;
> - it does not require an outer IPF loop, which would prevent error accumulation.
>
> On the other hand, our proposed algorithms also have some advantages:
> - they allow off-policy training, which is essential for preventing mode collapse (while [2] requires drawing samples from the current sampler for training);
> - they do not require access to the gradient of the target energy (while [2] uses it to compute the regression target);
> - they apply to non-Gaussian source distributions (while [2] considers only Gaussian-to-energy problems).
>
> This discussion will be included in the paper. We will also aim to include comparisons with [2] -- using a reimplementation or their published code, if it becomes available -- in the camera-ready version.
>
> ### KL and Wasserstein comparison (W3)
>
> The arrow is simply a typo in our table: certainly, lower KL is better. Thank you for the careful reading.
>
> Why trainable variance improves the Wasserstein but worsens the path KL is an interesting question. When we do not train variances, we achieve a worse fit to the target distribution (higher Wasserstein), that is, achieve worse satisfaction of the constraint in the optimisation problem defining the Schrödinger bridge -- recall that **the SB is the process with minimal cost (path KL) under the constraint of transporting $p_0$ to $p_1$ (that is, zero $\mathcal{W}_2^2$)**. Neural approximations do not meet this constraint perfectly, and we must study the tradeoff between target fit and path KL. For example, the reference process itself would have very bad $\mathcal{W}_2^2$, but zero path KL by construction. On the other hand, a model that transforms the source to the target by some arbitrary mapping (for example, a composition of a diffusion process transforming $p_0$ to noise and a reverse diffusion process transforming noise to $p_1$) could have low $\mathcal{W}_2^2$ but a high path KL.

---

> > ### Author Response · Authors · 2025-11-26
> >
> > ### Why GMM to GMM for energy-to-energy? (Q1)
> >
> > This is done simply to present a more challenging setting. Studying Gaussian-to-GMM in the energy-to-energy setting would be quite "easy", in particular because in the energy-to-energy setting, replay buffers of approximate samples for both $p_0$ and $p_1$ are kept. The buffers are always initialised with Gaussian samples across all experiments, and thus the replay buffer would already be initialised with perfect samples from $p_0$.
> >
> > **Thank you again for your careful reading and good questions. We hope the answers above have resolved your concerns.**

---

### Official Review · Reviewer_kkJj · 2025-11-04

**Soundness:** 3
**Presentation:** 3
**Contribution:** 3
**Rating:** 6
**Confidence:** 3

**Summary:**

The authors propose a modification of the Iterative Proportional Fitting (IPF) procedure to estimate a Schrödinger bridge between two distributions. Their approach does not require samples from either distribution, but instead relies on their unnormalized densities. At each step, they sample from the current approximation and add these samples to a replay buffer, similar to how it is done in reinforcement learning settings. They also explore the use of a learned diffusion scale coefficient in their framework and argue that this modification can improve any existing IPF algorithm. Finally, the authors demonstrate the performance of their proposed algorithm using synthetic toy datasets of Gaussian mixtures, as well as reward-driven energy estimations in the latent spaces of small generative models (GANs and Normalizing Flows), supporting potential applications to Style Transfer tasks.

**Strengths:**

* The idea is novel and natural, combining established concepts from Schrödinger bridges and entropy-regularized reinforcement learning. The choice of experiments to support the plausibility of the algorithm is adequate, though limited to toy examples.
* The theoretical foundations are well established, with an extensive literature review and a thorough background section on stochastic optimal transport and diffusion-based sampling. The appendix, which explains the metrics, is also well written.
* The paper is clearly written and easy to follow, although the experiment section is somewhat compressed in favor of the background overview.
* The paper demonstrates how the method could be applied to Style Transfer, but its main significance, in my view, lies in the thorough examination of the mathematical foundations.

**Weaknesses:**

* There are many unexplored applications for this method that would be more interesting to see in the paper. In particular, areas where unnormalized density is the primary source of information and data samples are very limited or nearly nonexistent—such as transferring molecular structures where energy or reward is given by physical simulation, or transferring grasping poses between different robot embodiments given a 3D object description. These examples would be better testbeds for the method’s applicability than outsourced image generation, as in the latter case it is difficult to distinguish between failure modes of latent-to-image generators and the method itself. Additionally, the source distribution in image generation is typically chosen to be Gaussian, which is again too simple and narrow for a comprehensive experimental overview. The authors do, however, mention in the conclusion that such experiments could be explored in future work.
* The FID discrepancy between rejection sampling and Schrödinger bridge samples is not well motivated and feels somewhat misleading, especially given the claim that the target distribution learned by the generative model is "ground truth." In reality, this may simply be a poor approximation.
* There are good analytical estimations in the literature for Schrödinger bridges between Gaussian mixtures, and for some reason, I did not find a measurement of the discrepancy between the learned bridge and the analytical approximation in these toy experiments.

**Questions:**

* Please try to examine and explain the findings regarding the FID discrepancy between Schrödinger bridge and rejection sampling in more detail.
* The paper could benefit from including more challenging distribution pairs, even in the toy setting, such as the widely adopted moons and swiss roll datasets, as well as higher-dimensional Gaussian mixtures.

---

> ### Author Response · Authors · 2025-11-26
>
> Thank you for your review. We appreciate that you describe the proposed data-to-energy SB algorithm as novel, natural, and theoretically grounded and that you found the paper easy to read. We will now proceed to respond to the weaknesses and questions.
>
> ### Many unexplored applications (W1)
>
> We agree that dynamic measure transport algorithms that do not rely on samples from the target distribution open the door to many unexplored applications.
> However, as you noted in the strengths, the main purpose of this paper is to present the first general algorithm for Schrödinger bridges between two general distributions, where one or both have no samples available.
>
> To expand on the style transfer experiments, where a prior is transported to a posterior in latent space, we show an example of learning an energy-to-energy bridge in the latent space of SN-GAN, transporting the posterior distribution given the class label "dog" to that given "cat", under a pretrained classifier and with no data samples from either class. Images are shown in Appendix F.3 in the update manuscript.
>
> ### FID not motivated (W2, Q1)
>
> Because we are performing translation in the **latent space** of a GAN without data, evaluation of a sampler against **images** of the target class will incur bias from two sources:
> - from an imperfect prior/decoder: the distribution of decoded prior samples does not match the distribution of real images
> - from an imperfect classifier: the distribution of images weighted by the classifier's probability of a class does not match the distribution of real images of that class.
>
> To better place our FID results in context, for the outsourced Schrödinger bridge experiment, we compute FID between a set of ground truth images of the target class and:
> - an independent sample of the same size of images of the same class (which would give 0 in the limit of infinite sample size);
> - samples from our trained Schrödinger bridge model;
> - samples from a diffusion sampler in latent space that aims to sample the same target distribution as the bridge; this follows [Venkatraman et al., "Outsourced diffusion sampling...", ICML'25];
> - decoded samples from the *exact* posterior in latent space, obtained via rejection sampling;
> - samples obtained by Langevin dynamics in the latent space, again asymptotically sampling from the same target density.
>
> **FID score for SN-GAN trained on CIFAR-10.**
>
> |     Algorithm    |  Car   |  Cat   |  Dog   |  Horse |
> | :--------------: | :----: | :----: | :----: | :----: |
> |Same class        | 10.371 | 17.950 | 15.034 | 12.876 |
> |Rejection Sampling| 31.334 | 42.242 | 43.691 | 35.019 |
> |Langevin          | 40.302 | 45.949 | 42.736 | 34.873 |
> |Diffusion Sampler | 83.940 |  53.253   | 60.512 |   57.251   |
> |Outsourced SB     | 22.312	| 40.489 | 37.287 | 33.021 |
>
> (Some results are still running and will be updated here by 27 November AOE.)
>
> **FID score for StyleGAN trained on CIFAR-10.**
>
> |     Algorithm    | Horse  | Truck  |
> | :--------------: | :----: | :----: |
> |Same class        | 12.876 |  9.289 |
> |Rejection Sampling| 97.855 | 76.403 |
> |Langevin          | 85.611 | 68.678 |
> |Outsourced SB     | 58.988	| 55.346 |
>
> **Among amortised and approximate sampling methods (Langevin, diffusion sampler, our SB), the SB achieves the lowest FID.**
>
> We recognise that FID does not perfectly capture distribution fit in the latent space: indeed, we see decoded bridge samples having lower FID than exact ones.
>
> ### Comparison with analytical estimations (W3, Q2)
>
> As far as we are aware, the Schrödinger bridge between Gaussian mixtures (suggested in your comment) does not admit a closed form. However, the bridge between two arbitrary Gaussians is known analytically [Bunne et al., "The Schrödinger bridge between Gaussian measures...", AISTATS'23]. To study the quality of the bridges learnt by different algorithms, we perform the following experiment: we fit bridges between two two-dimensional Gaussians $\mathcal{N}(0,I)$ and $\mathcal{N}\left(\begin{bmatrix} 1 \\ 1 \end{bmatrix},\begin{bmatrix} 3.0, 1.2 \\ 1.2, 1.0 \end{bmatrix}\right)$.
>  We then measure the Wasserstein distance between marginal samples from the learnt and true bridges at each intermediate time point.
>
> Figure 12 in the revised pdf shows the results. We see that **our data-to-energy SB matches the target distribution (W2 approaches the self-W2 between two samples from the target at the final time endpoint), but also closely matches the analytical marginals at intermediate times -- better so than the SB trained by data-to-data IPF**, as well as DSBM, and the diffusion sampler, which has no goal of matching the intermediate distributions of the Schrödinger bridge.
>
> We hope this result lends more support to our claims.
>
> **We would like to once again thank the reviewer for the supportive comments on our paper. We appreciate the careful reading of the paper and value the important issues you have raised.**

---

### Meta-Review · Area_Chair_fUEA · 2026-01-03

**Summary:**

The paper addresses an important and relatively underexplored setting for Schrödinger bridge learning, namely when one or both endpoint distributions are specified only through unnormalised densities rather than samples, and proposes a data-to-energy (and energy-to-energy) IPF procedure based on a VarGrad, off-policy objective. The contributions are technically meaningful, including a generalisation of IPF to sample-free endpoints, a practical training scheme that avoids target energy gradients, and evidence that learning the diffusion coefficient can improve discretised SB training and enable new applications such as latent-space posterior sampling and image-to-image translation without target samples. The reviewers raised several substantive concerns, but these were largely addressed to a satisfactory degree in the rebuttal. First, questions about evaluation and positioning, particularly around FID comparisons and missing baselines, were mitigated by adding broader multi-baseline FID tables, clarifying the limitations of FID in this setting, and strengthening toy experiments with an analytic Gaussian bridge comparison that directly measures intermediate-time accuracy. Second, while convergence guarantees for the proposed off-policy IPF remain conditional, the authors provided a clear argument situating their method within standard discrete-time IPF assumptions commonly adopted in neural SB work, which is reasonable for the intended scope and aligns with existing practice. Third, concerns about scalability and computational cost were addressed with concrete timing breakdowns on modern hardware, a clear explanation of why trajectory-level costs are unavoidable in data-to-energy settings, and a convincing case that off-policy VarGrad meaningfully reduces backpropagation overhead compared to naive alternatives. Finally, several sources of confusion and presentation issues (such as the Path KL typo, the role of Langevin noise, and the distinction between terminal buffers and SDE noise) were convincingly clarified with added explanations and ablations. While some gaps remain, notably the lack of direct empirical comparison to certain concurrent SB variants and the reliance on pretrained generators for high-dimensional image experiments, these limitations are acknowledged and do not undermine the core contribution. Overall, the paper presents a solid and novel extension of Schrödinger bridge methodology to a practically relevant and technically challenging regime, the authors responded constructively and substantively to reviewer feedback, and the work meets the bar for acceptance.

**Reviewer Concerns:**

Most of the major concerns raised during review were addressed in the rebuttal. Evaluation-related issues, including the motivation and interpretation of FID, missing baselines, and the apparent mismatch between latent-space objectives and image-level metrics, were substantially clarified through additional experiments, expanded FID tables, and explicit discussion of bias sources. Questions about theoretical soundness and convergence were not resolved in a formal sense, but the authors provided a coherent justification grounded in standard IPF assumptions used across the neural Schrödinger bridge literature, which sufficiently addresses the intent of those critiques. Concerns about computational efficiency and scalability were directly answered with detailed runtime and memory analyses and a principled explanation of why such costs are intrinsic to the data-to-energy setting. Remaining concerns include the absence of direct experimental comparisons with some concurrent methods and the dependence on pretrained generative models for image experiments, which remain open but were clearly acknowledged and contextualised.

**Reviewer Scores:**

- Reviewer kkJj (initial score: 6): Likely a slight increase to a stronger accept, given that their main questions about FID interpretation, additional toy experiments, and analytical comparisons were explicitly answered and expanded upon.
- Reviewer Rvjv (initial score: 6): Likely unchanged or marginally increased, as concerns about convergence, Path KL interpretation, and methodological positioning were carefully addressed, including acknowledgement of limitations and correction of presentation errors.
- Reviewer k5Aj (initial score: 4): Likely unchanged. Although many misunderstandings were clarified in detail, the reviewer’s skepticism about novelty, experimental design, and relevance of the problem setting suggests limited receptiveness to the rebuttal.
- Reviewer oxUm (initial score: 4): Likely unchanged. While timing and scalability questions were answered concretely, the reviewer’s broader doubts about impact and reliance on pretrained models were only partially mitigated.

---

### Decision · Program_Chairs · 2026-01-26

Accept (Poster)